# DOPPLER: Differentially Private Optimizers with Low-pass Filter for Privacy Noise Reduction

**Xinwei Zhang**[*]
University of Southern California

**Zhiqi Bu**[†]
Amazon

**Mingyi Hong**
University of Minnesota

**Meisam Razaviyayn**[*]
University of Southern California

## Abstract

Privacy is a growing concern in modern deep-learning systems and applications. Differentially private (DP) training prevents the leakage of sensitive information in the collected training data from the trained machine learning models. DP optimizers, including DP stochastic gradient descent (DPSGD) and its variants, privatize the training procedure by gradient clipping and *DP noise* injection. However, in practice, DP models trained using DPSGD and its variants often suffer from significant model performance degradation. Such degradation prevents the application of DP optimization in many key tasks, such as foundation model pre-training. In this paper, we provide a novel *signal processing perspective* to the design and analysis of DP optimizers. We show that a "frequency domain" operation called *low-pass filtering* can be used to effectively reduce the impact of DP noise. More specifically, by defining the "frequency domain" for both the gradient and differential privacy (DP) noise, we have developed a new component, called DOPPLER. This component is designed for DP algorithms and works by effectively amplifying the gradient while suppressing DP noise within this frequency domain. As a result, it maintains privacy guarantees and enhances the quality of the DP-protected model. Our experiments show that the proposed DP optimizers with a low-pass filter outperform their counterparts without the filter by $3\% - 10\%$ in test accuracy on various models and datasets. Both theoretical and practical evidence suggest that the DOPPLER is effective in closing the gap between DP and non-DP training.

## 1 Introduction

A rapidly growing number of modern machine learning applications in computer vision, natural language processing, and their mixtures rely on the development of large foundation models, whose performance heavily depends on the huge amounts of data collected from individual users. The leakage of potentially sensitive information in training data has become an increasingly critical issue when releasing and using machine learning models. Unfortunately, modern complex models have a strong ability to memorize the exact training data during the training processing Carlini et al. (2021); Pan et al. (2020). To alleviate the possible privacy leakage in the model training procedure, privacy-preserving optimization has attracted both researchers' and practitioners' interests.

Differential Privacy (DP) Dwork and Roth (2014) provides a strong theoretical guarantee with an easy and nearly plug-and-play mechanism, i.e., gradient clipping and noise injection, for existing optimization algorithms to guarantee the privacy of training procedures Abadi et al. (2016). By directly applying the DP mechanism to existing optimizers, DP optimizers have achieved decent performance in fine-tuning foundation models (Yu et al., 2021; Bu et al., 2024) or training small

---

[*]`xinweiz,razaviya@usc.edu`, [†]This work is not affiliated with Zhiqi Bu's position at Amazon.
38th Conference on Neural Information Processing Systems (NeurIPS 2024).

models (De et al., 2022). However, the performance of the pretraining tasks and training large foundation models using DP optimizers still remain unsatisfactory. This is because, as the DP theory suggests, the amount of injected DP noise is proportional to the number of model parameters and the total update steps (Abadi et al., 2016). Thus, the performance of large foundation models trained with DP optimizers degrades severely. To put it in perspective, pretraining a foundation model for an image classification task on the CIFAR dataset from randomly initialized weights takes around 100 epochs, with $300K$ steps (Dosovitskiy et al., 2020); the pretraining of BERT for natural language processing task takes $1M$ steps (Devlin et al., 2019), and pretraining LLAMA takes more than $250K$ steps (Touvron et al., 2023). The number of trainable parameters is also huge for these tasks, ranging from $300M$ to $70B$. Therefore, the huge amount of injected DP noise severely degrades the performance of the final model trained with DP optimizers.

To improve the performance of DP optimizers, existing research takes two approaches: 1) designing models that are less sensitive to DP noise, e.g., using group normalization, weight standardization, weight smoothing, and smooth activation layers (De et al., 2022; Papernot et al., 2021; Wang et al., 2020), and 2) designing adaptive DP optimizers that inject *relatively* smaller noise, e.g., adaptive clipping, sparse gradient, and dynamic privacy budget (Andrew et al., 2021; Yu et al., 2021; Luo et al., 2021; Hong et al., 2022). However, existing methods only work for certain models and tasks at the cost of consuming more privacy budget. Moreover, most of the methods only demonstrate empirical improvements and do not provide theoretical justification for improving DP optimization. Therefore, there is a strong need for an approach that improves the performance of DP optimizers, which has the following properties: 1) has a solid theoretical guarantee, 2) is easy to implement, and 3) is compatible with most existing DP optimization improving methods.

Motivated by the above needs, in this paper, we develop a module that can easily be integrated into the DP training optimizers. We provide both theoretical and empirical justification for our proposed module. Specifically, our **contributions** are as follows:

- **Frequency-domain analysis:** We introduce the notion of frequency domain analysis on (DP) optimizers. This analysis sheds light on how "noise" affects the "signal" part of the update directions viewed as a sequence of update steps, rather than independent update steps.

- **Low-pass filter approach:** Based on our frequency-domain analysis, we propose a low-pass filtering approach named DOPPLER, that post-processes the privatized gradient to reduce DP noise and improve DP optimizers' performance. Our low-pass filter reduces noise in the frequency domain, which is orthogonal to existing DP noise reduction approaches in the time domain and, therefore, can be easily combined with other existing techniques to further reduce noise.

- **Theoretical Analysis:** We provide a novel theoretical analysis for the proposed low-pass filtering approach. Specifically, by introducing certain frequency domain assumptions on the gradients, we provide the convergence and privacy guarantee for DPSGD with the low-pass filter. Unlike existing methods that involve trading off noise with bias (e.g., adaptive clipping), or based on approximation (e.g., low-rank decomposition), our proposed algorithm does not introduce extra bias, model modification, or extra privacy cost.

- **Numerical results:** Our extensive numerical experiments compare the performance of a variety of DP optimizers with and without the low-pass filter on different models and datasets. Our results show that DP optimizers equipped with the low-pass filter outperform the ones without the filter.

## 2 Preliminaries

In this section, we discuss notations, assumptions, and some related prior work on DP optimization:

### 2.1 Notations & assumptions

In this paper, we aim to optimize the Empirical Risk Minimization (ERM) problem:

$$\min_{\mathbf{x} \in \mathbb{R}^d} F(\mathbf{x}), \quad \text{where} \ \ F(\mathbf{x}) := \frac{1}{N} \sum_{i=1}^{N} f(\mathbf{x}; \xi_i). \tag{1}$$

Here, $\mathcal{D} = \{\xi_i\}_{i=1}^{N}$ is the training dataset with $N$ samples. Further denote the lower bound of the problem as $f^\star = \inf f(\mathbf{x})$. Throughout the paper, we use $(\cdot)_t$ to denote the update steps, $\mathcal{N}(\mu, \sigma^2)$

to denote the Gaussian distribution with mean $\mu$ and variance $\sigma^2$. We also assume the problem (1) satisfies the following assumptions.

**A 1 (Smoothness)** $F(\cdot)$ *is L-smooth, i.e.,* $\|\nabla F(\mathbf{x}) - \nabla F(\mathbf{y})\| \le L \|\mathbf{x} - \mathbf{y}\|, \ \forall \mathbf{x}, \mathbf{y} \in \mathbb{R}^d$.

**A 2 (Bounded Variance)** *The per-sample gradient has bounded variance, i.e.,*

$$\mathbb{E}_{\xi \in \mathcal{D}} \|\nabla f(\mathbf{x}; \xi) - \nabla F(\mathbf{x})\|^2 \le \sigma_{SGD}^2, \ \forall \mathbf{x} \in \mathbb{R}^d.$$

**A 3 (Bounded Gradient)** *The per-sample gradient has a bounded norm, i.e.,*

$$\|\nabla f(\mathbf{x}; \xi)\| \le G, \ \forall \mathbf{x} \in \mathbb{R}^d, \xi \in \mathcal{D}.$$

Let us briefly comment on these assumptions: A1 and A2 are standard in non-convex optimization (Allen-Zhu and Hazan, 2016; Zaheer et al., 2018; Abadi et al., 2016); and A3 is commonly used in analyzing the convergence of DP algorithms (Abadi et al., 2016; Wang et al., 2020; Andrew et al., 2021) to avoid introducing the clipping bias. Since the impact of clipping is not the major focus of this paper, we follow the existing analyses and use A3 to simplify our theoretical analysis.

### 2.2 Differential privacy (DP) and differentially private SGD (DPSGD)

Differential privacy is a gold standard of privacy to protect the privacy of individuals:

**Definition 1 (($\epsilon, \delta$)-DP (Dwork and Roth, 2014))** *A randomized mechanism $\mathcal{M}$ is said to be $(\epsilon, \delta)$-differentially private, if for any two neighboring datasets $\mathcal{D}, \mathcal{D}'$ ($\mathcal{D}, \mathcal{D}'$ differ only by one sample) and for any measurable output set $\mathcal{S}$, it holds that $\Pr[\mathcal{M}(\mathcal{D}) \in \mathcal{S}] \le \mathrm{e}^\epsilon \Pr[\mathcal{M}(\mathcal{D}') \in \mathcal{S}] + \delta$.*

A popular practical differentially private approach to finding an (approximate) solution to the ERM optimization problem (1) is Differentially Private Stochastic Gradient Descent (DPSGD) (Abadi et al., 2016) and its variants, including DP-Adam and DP-Lora (Yu et al., 2021). To protect DP, DPSGD considers applying the commonly used Gaussian mechanism (Dwork and Roth, 2014; Abadi et al., 2016) at each iteration of the stochastic gradient descent method. The Gaussian mechanism provides a DP guarantee by injecting additive noise into the algorithm output.

**Definition 2 (Gaussian Mechanism (Dwork and Roth, 2014; Zhao et al., 2019))** *Suppose an algorithm $\mathcal{A} : \mathcal{D} \rightarrow \mathbb{R}^d$ has $\ell_2$ sensitivity $\Delta_{\mathcal{A}}$, i.e., $\max_{\mathcal{D}, \mathcal{D}'} \|\mathcal{A}(\mathcal{D}) - \mathcal{A}(\mathcal{D}')\| \le \Delta_{\mathcal{A}}$. Then, for any $\epsilon > 0$ and $\delta \le 0.05$, by adding random Gaussian noise to the output of the algorithm $M(x) = \mathcal{A}(x) + \mathbf{w}$, with $\mathbf{w} \sim \mathcal{N}(0, \sigma_{\mathrm{DP}}^2 I_d)$, where $\sigma_{DP} = \frac{\Delta_{\mathcal{A}} \sqrt{2 \ln(1/2\delta)}}{\epsilon} + \frac{\Delta_{\mathcal{A}}}{\sqrt{2\epsilon}}$, the algorithm $M$ is $(\epsilon, \delta)$-DP.*

The DPSGD algorithm, presented in Algorithm 1, first samples a mini-batch $\mathcal{B}^t$ of size $B$ and computes the per-sample gradient at each step $t$. Then, it applies the Gaussian mechanism by clipping the per-sample gradient and injecting DP noise. The clipping operation bounds the sensitivity of the stochastic gradients to $C$, e.g., $\mathrm{clip}\,(\nabla f, C) = \min \left\{1, \frac{C}{\|\nabla f\|}\right\} \nabla f$ or $\frac{C}{\|\nabla f\|} \nabla f$. Finally, the

---
**Algorithm 1** DPSGD algorithm

---
**Input:** $\mathbf{x}_0, \mathcal{D}, C, \eta, \sigma_{\mathrm{DP}}$
**for** $t = 0, \dots, T - 1$ **do**
     Uniformly draw minibatch $\mathcal{B}_t$ from $\mathcal{D}$
     $\mathbf{g}_t = \frac{1}{B} \sum_{\xi_i \in \mathcal{B}_t} \mathrm{clip}\,(\nabla f(\mathbf{x}_t; \xi_i), C) + \mathbf{w}_t$
         where $\mathbf{w}_t \sim \mathcal{N}(0, \sigma_{\mathrm{DP}}^2 \cdot \mathbf{I}_d)$
     $\mathbf{x}_{t+1} = \mathbf{x}_t - \eta_t \mathbf{g}_t,$
**end for**

---

algorithm updates the model parameter with the privatized mini-batch gradient. It has been shown that DPSGD guarantees $(\epsilon, \delta)$-DP with sufficiently large injected noise (Abadi et al., 2016).

**Theorem 1 (Privacy Guarantee (Abadi et al., 2016))** *Given $N, B, T$ and $C$, there exist positive constants $u, v$, such that for any $\epsilon < \frac{uB^2T}{N^2}, \delta > 0$, by choosing $\sigma_{\mathrm{DP}}^2 \ge v \frac{C^2 T \ln(\frac{1}{\delta})}{N^2 \epsilon^2}$, Algorithm 1 is guaranteed to be $(\epsilon, \delta)$-DP.*

### 2.3 Related work

**Effective DP training:** Improving the performance of DP training has been widely studied. Adaptive gradient clipping (Andrew et al., 2021) estimates the size of the gradient privately and adaptively

changes the clipping threshold to avoid injecting large DP noise; automatic clipping (Bu et al., 2024) replaces the clipping operation with normalization to avoid injecting large DP noise when the gradient becomes small; Hong et al. (2022) proposes using a time-varying privacy budget at each step which injects non-static DP noise based on the gradient to reduce the impact of the DP noise. As the injected DP noise variance scales with the model size, reducing the number of trainable parameters with adapters, low-rank weights, or quantized models has also been used to reduce the DP noise magnitude (Yu et al., 2021; Luo et al., 2021; Yu et al., 2021). De et al. (2022); Papernot et al. (2021); Wang et al. (2020) use special model structures that are less sensitive to DP noise, including group normalization, weight standardization, smoothed activation, and smoothed weights.

These methods aim to reduce the magnitude of the injected DP noise or make the model and/or the DP algorithm less sensitive to large DP noise. However, the improvement is either empirical or only works for specific model structures and is unable to be generalized to other DP training tasks.

**Signal processing for optimization:** A few existing works analyze the optimization procedure from the signal processing perspective. They mainly focus on optimizing strongly convex problems using deterministic algorithms (Hu and Lessard, 2017; An et al., 2018); Gannot (2022) provides stability and convergence analysis from the frequency domain for inexact gradient methods. However, the results are still restricted to non-DP optimization and to strongly convex problems.

# 3 A signal processing perspective

As discussed in Section 2.3, most of the existing works that aim to improve the performance of DP training are reducing the *per-iteration* injected DP noise. These approaches treat the update directions in each step independently, omitting the underlying dynamics and correlations between the steps. However, the gradient directions typically change smoothly due to the smoothness of the machine-learning model; therefore, the update directions are not independent over time.

With the intuition that the gradients over iterations are not independent, we provide the *frequency-domain* analysis of the stochastic gradients. Specifically, in the frequency domain, we treat the (stochastic) gradients from $t = 0$ to $t = T$ as a time series and analyze the long-term correlation and dependencies across all gradients. In contrast, the time domain refers to the analyses that only focus on the gradient at step $t$: for instance, Bu et al. (2024); Yang et al. (2022) leverage the $L$-Lipschitz smoothness or the second-order Taylor expansion to bound/approximate the objective function in step $t$ after the update is performed. By analyzing the DP optimizers' updates in the frequency domain, we can make use of our prior knowledge of the correlation among the update directions.

To explain our motivation in a simplified manner, we temporarily ignore the per-sample gradient clipping and focus our narrative on the noise: suppose $\mathbf{x}_{t+1} = \mathbf{x}_t - \eta_t \mathbf{g}_t$ with $\mathbf{g}_t = \nabla F(\mathbf{x}_t) + \mathbf{w}_t$ and $\mathbf{w}_t$ being the Gaussian noise. We will come back to the clipping in Section 3.2. In what follows, we decompose the sequence of stochastic gradients $\{\mathbf{g}_t\}$ into two parts: 1) the gradient signal $\{\nabla F(\mathbf{x}_t)\}$ and the noise $\{\mathbf{g}_t - \nabla F(\mathbf{x}_t)\}$. We employ the power spectral density (PSD) to characterize the distribution of power into frequency components of a continuous signal. Mathematically, the power spectral density (PSD) of a sequence $\{s_t\}_{t=0,1,\dots}$ is $P_s(\nu) = \mathcal{F}\{\phi_s(\tau)\}$, where $\mathcal{F}$ denotes the Fourier transform from time domain ($\tau$) to frequency domain ($\nu$) (Oppenheim et al., 1996), and $\phi$ is the auto-correlation coefficient as $\phi_s(\tau) = \mathbb{E}\langle s_t, s_{t-\tau}\rangle$.

On the one hand, the gradient sequence $\{\nabla F(\mathbf{x}_t)\}_{t=0,1,\dots}$ can be treated as a low-frequency signal, where we apply the Cauchy Schwarz inequality to get

$$\phi_{\nabla f}(\tau) = \mathbb{E}\langle \nabla F(\mathbf{x}_t), \nabla F(\mathbf{x}_{t-\tau})\rangle = \frac{1}{2}\mathbb{E}\left[\|\nabla F(\mathbf{x}_t)\|^2 + \|\nabla F(\mathbf{x}_{t-\tau})\|^2 - \|\nabla F(\mathbf{x}_t) - \nabla F(\mathbf{x}_{t-\tau})\|^2\right]$$

$$\geq \frac{1}{2}\mathbb{E}\left[\|\nabla F(\mathbf{x}_t)\|^2 + \|\nabla F(\mathbf{x}_{t-\tau})\|^2 - L^2\eta^2\tau\sum_{i=1}^{\tau}\|\mathbf{g}_{t-i}\|^2\right].$$

This indicates that as long as the *stepsize $\eta$ is small*, the auto-correlation coefficients decrease as $\tau$ increases (as illustrated in Figure 1a, blue line). Therefore, the PSD also decreases as $\nu$ increases, i.e., $\{\nabla F(\mathbf{x}_t)\}$ is a low-frequency signal (as illustrated in Figure 1b, blue line).

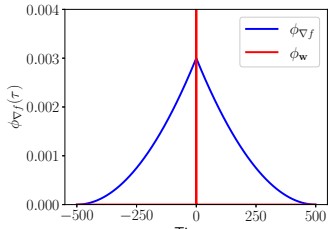
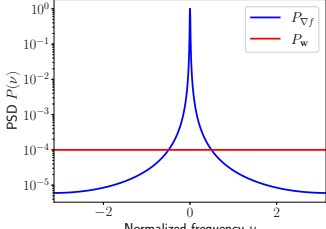
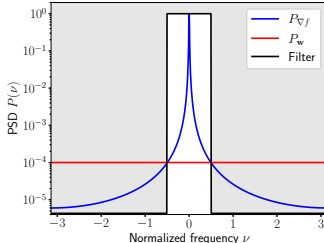

| (a) Auto-correlation coefficients | (b) PSD coefficients | (c) PSD & an ideal low-pass filter |

Figure 1: An illustration of the auto-correlation $\phi(\tau)$ and power spectrum density $P(\nu)$ of $\{\nabla F(\mathbf{x}^t)\}$ and $\mathbf{w}^t$ where $\phi_{\nabla f}$ decays proportional to $\tau^2$ and $\mathbf{w}_t$ is a white noise. (c) illustrates how an ideal low-pass filters out the high-frequency noise and keeps the low-frequency signal.

On the other hand, the noise signal $\{\mathbf{w}_t\} := \{\mathbf{g}_t - \nabla F(\mathbf{x}_t)\}$ is a white noise, where its auto-correlation is non-zero when $\tau = 0$ and is zero otherwise (as illustrated in Figure 1a, red line):

$$\phi_{\mathbf{w}}(\tau) = \mathbb{E}\left\langle \mathbf{w}_t, \mathbf{w}_{t-\tau} \right\rangle \begin{cases} \leq d\sigma_{\mathrm{DP}}^2 + \frac{\sigma_{\mathrm{SGD}}^2}{B}, & \tau = 0 \\ = 0, & \text{otherwise.} \end{cases}$$

Therefore, $P_{\mathbf{w}}(\nu) = d\sigma_{\mathrm{DP}}^2 + \frac{\sigma_{\mathrm{SGD}}^2}{B}, \forall \nu$ (as illustrated in Figure 1b, red line).

### 3.1 Low-pass filter and noise reduction

From the above discussion, we observed that although the gradient and the DP noise are not separable in each step $t$ (time-domain), they are distinguishable in the *frequency domain*. In particular, *the noise power is equally distributed over all frequencies, while the gradient is concentrated around the lower frequencies.* Therefore, we can apply the classical signal processing tools, such as frequency domain low-pass filters, to help improve the performance of DP optimization.

A low-pass filter amplifies the low-frequency component of the signal and suppresses the high-frequency part. Figure 1c shows an ideal low-pass filter that keeps the frequencies where the gradient is larger ($\nu \in [-0.6, 0.6]$) and blocks the frequencies where the noise is larger ($\nu < -0.6 \cup \nu > 0.6$). Signal-to-Noise Ratio (SNR) is a useful measure to characterize the quality of a noisy signal, i.e., the privatized gradient $\{\mathbf{g}_t\}$ in DP optimization. Given the PSD of the gradient and the noise, the SNR of the privatized gradient is $\frac{\sum_\nu P_{\nabla f}(\nu)}{\sum_\nu P_{\mathbf{w}}(\nu)}$. As illustrated in Figure 1, when there is no low-pass filter (i.e., in Figure 1b), the SNR is small as the noise dominates in the high-frequency. In contrast, by applying the low-pass filter (i.e., in Figure 1c), most of the signals in the low-frequencies are kept, and the noise in the high frequencies is filtered, so the SNR increases. A linear low-pass filter on $\{\mathbf{g}_t\}$ can be written as a recursive linear combination of the history signals:

$$\mathbf{m}_t = -\sum_{\tau=1}^{n_a} a_\tau \mathbf{m}_{t-\tau} + \sum_{\tau=0}^{n_b} b_\tau \mathbf{g}_{t-\tau},$$

where the sequence $\{\mathbf{m}_t\}$ is the filtered output, $\{a_\tau\}, \{b_\tau\}$ are the filter coefficients. Additionally, the "order" of the filter is defined as $\max\{n_a, n_b\}$. By carefully designing the coefficients, the low-pass filter can take different shapes and filter different frequencies.

In contrast to low-pass filters, the existing approaches improving the performance of DP optimization can be viewed as increasing the SNR in the time domain, i.e., reducing the magnitude of the noise injected in each step while preserving most of the gradient signal. Because the low-pass filter reduces noise in the frequency domain, and the existing noise reduction approaches lie in the time domain, the two approaches are orthogonal to each other. Therefore, the low-pass filter can be combined with existing approaches to further improve the DP optimizers' performance.

### 3.2 The impact of per-sample gradient clipping

The above analysis assumes that the clipping operation is inactive by choosing a large enough clipping threshold $C$. In practice, the clipping operation is usually active. By assuming the clipped

gradient $\nabla F_C(\mathbf{x})$ has zero curl, the DP optimizer optimizes an alternative problem:

$$\min_{\mathbf{x} \in \mathbb{R}^d} F_C(\mathbf{x}), \text{ where } F_C(\mathbf{x}) = \int_0^1 \nabla F_C(z\mathbf{x})^\top \mathbf{x} \mathrm{d}z, \ \nabla F_C(\mathbf{x}) = \frac{1}{N} \sum_{\xi \in \mathcal{D}} \mathrm{clip}\left(\nabla f(\mathbf{x}; \xi), C\right). \quad (2)$$

Then the signal of the DP optimizer becomes the gradient of the alternative problem $\{\nabla F_C(\mathbf{x}_t)\}$ and the noise becomes $\{\mathbf{w}_t\} = \{\mathbf{g}_t - \nabla F_C(\mathbf{x}_t)\}$. As clipping is a non-expansive operator, i.e., $\|\mathrm{clip}\left(\mathbf{x}, C\right) - \mathrm{clip}\left(\mathbf{y}, C\right)\| \leq \|\mathbf{x} - \mathbf{y}\|$, the alternative problem $F_C(\cdot)$ is also $L'$-smooth with $L' \leq L$. Therefore, a similar argument could be made on the gradient signal $\{\nabla F_C(\mathbf{x}_t)\}$ and the noise $\{\mathbf{w}_t\}\}$ when the clipping threshold is small and the clipping operation is active.

# 4   The proposed DOPPLER approach

Building on the discussions in Section 3, we proposed a universal approach to improve DP optimization performance: **DP OP**timizer with Low-**P**ass fi**L**ter for nois**E R**eduction (**DOPPLER**). Taking DPSGD as an example, by applying DOPPLER, the main steps of the modified DPSGD algorithm are illustrated in Algorithm 2. The key steps of the low-pass filter are described in Lines 6-8. Line 6 computes the filtered update direction $\mathbf{m}_t$ as a recursive linear combination of the current gradient, past gradients, and past update directions. $\mathbf{m}_t$ estimates the first moment of the privatized gradient and can be expanded as a moving average of $\mathbf{g}_t$, i.e., $\mathbf{m}_t = \sum_{\tau=0}^{t+n_b} \kappa_\tau \mathbf{g}_{t-\tau}$. However, as $\{\mathbf{m}_\tau\}, \{\mathbf{g}_\tau\}$ are initialized with zeros (Line 2), $\mathbf{m}_t$ is biased towards zero, especially in the early steps. To correct the initialization bias, in Line 7, the optimizer computes the bias correction factor $c_{a,t}$ that is used in Line 8 to guarantee the weights $\kappa_\tau$ in the moving average are summed to 1.

**Connection to momentum method:** The DOPPLER approach is a generalized version of the momentum method from a first-order filter to higher orders. The momentum method uses *one* buffer $\mathbf{m}_t$ to store the exponential moving average of $\mathbf{g}_t$, while DOPPLER uses *multiple* buffers $\{\mathbf{m}_{t-\tau}\}_{\tau=0}^{n_a-1}$ to compute a more complex moving average of $\mathbf{g}_t$.

**Compatibility:** Algorithm 2 demonstrates how DOPPLER can be combined with the DPSGD algorithm while it is not restricted to DPSGD. The DOPPLER approach is compatible with other advanced DP optimizers, e.g., Adam (Kingma and Ba, 2015; Tang et al., 2024) and GaLore (Zhao et al., 2024). It serves as a base component for DP optimizers to improve their performance.

# 5   Theoretical analysis

## 5.1   Convergence analysis

In this section, we analyze the convergence of DPSGD with DOPPLER. First, we make the following assumption on the gradient auto-correlation coefficients.

**A 4 (Gradient auto-correlation)** *For all $t \in \{0, \ldots, T-1\}$, there exists sequences $\{c_\tau\}, \{c_{-\tau}\}$ such that the following condition holds:*

$$\langle \nabla F(\mathbf{x}_t), \nabla F(\mathbf{x}_{t-\tau}) \rangle \geq c_\tau \|\nabla F(\mathbf{x}_t)\|^2 + c_{-\tau} \|\nabla F(\mathbf{x}_{t-\tau})\|^2, \quad \forall \tau \geq 0, \quad (3)$$

$$c_{-\tau} \geq 0, \quad \forall \tau \geq 0. \quad (4)$$

---
**Algorithm 2** DPSGD with DOPPLER
---
1: **Input:** $\mathbf{x}_0, \mathcal{D}, \eta, C, \sigma_{\mathrm{DP}}, \{a_\tau\}_{\tau=1}^{n_a}, \{b_\tau\}_{\tau=0}^{n_b}$
2: **Initialize:** $\{\mathbf{m}_{-\tau}\}_{\tau=1}^{n_a} = 0, \{\mathbf{g}_{-\tau}\}_{\tau=1}^{n_b} = 0, \{c_{a,-\tau}\}_{\tau=1}^{a_n} = 0, \{c_{b,-\tau}\}_{\tau=0}^{b_n} = 0$
3: **for** $t = 0, \ldots, T-1$ **do**
4:     Randomly draw minibatch $\mathcal{B}_t$ from $\mathcal{D}$
5:     $\mathbf{g}_t = \frac{1}{|\mathcal{B}_t|} \sum_{\xi \in \mathcal{B}_t} \mathrm{clip}\left(\nabla f(\mathbf{x}; \xi), C\right) + \mathbf{w}_t$               *# Compute private gradient*
              where $\mathbf{w}_t \sim \mathcal{N}(0, \sigma_{\mathrm{DP}}^2 \cdot \mathbf{I}_d)$
6:     $\mathbf{m}_t = -\sum_{\tau=1}^{n_a} a_\tau \mathbf{m}_{t-\tau} + \sum_{\tau=0}^{n_b} b_\tau \mathbf{g}_{t-\tau}$                          *# Apply filter*
7:     $c_{b,t} = 1, c_{a,t} = -\sum_{\tau=1}^{n_a} a_\tau c_{a,t-\tau} + \sum_{\tau=0}^{n_b} b_\tau c_{b,t-\tau}$     *# Compute bias*
8:     $\hat{\mathbf{m}}_t = \mathbf{m}_t / c_{a,t}$                                                      *# Correct initialization bias*
9:     $\mathbf{x}_{t+1} = \mathbf{x}_t - \eta \hat{\mathbf{m}}_t$                                             *# Parameter update*
10: **end for**
---

Clearly, we have $c_0 = \frac{1}{2} > 0$. From the discussion in Section 3, we see that the above assumption can be satisfied as long as $\eta$ is small enough, i.e.,

$$\eta \leq \frac{\sqrt{(1 - 2c_{-\tau}) \|\nabla F(\mathbf{x}_{t-\tau})\|^2 + (1 - 2c_\tau) \|\nabla F(\mathbf{x}_t)\|^2}}{L\sqrt{\|\sum_{\tau_1=1}^\tau \nabla F(\mathbf{x}_{t-\tau_1})\|^2 + \tau(d\sigma_{\text{DP}}^2 + \sigma_{\text{SGD}}^2/B)}} = \mathcal{O}\left(\sqrt{\frac{1}{\tau}}\right).$$

The pattern of the sequence $\{c_\tau\}$ characterizes the frequency of the gradients as discussed in Section 3. If $c_\tau$'s are all positive and slowly decreasing, then $\nabla F(\mathbf{x}_t)$ and $\nabla F(\mathbf{x}_{t-\tau})$ are highly correlated, so $\{\nabla F(\mathbf{x}_t)\}$ lies in lower frequencies. However, $c_\tau$ is not necessarily positive. When some of $c_\tau$'s are negative, or $c_\tau$'s are oscillating between positive and negative values, it means that $\nabla F(\mathbf{x}_t)$ and $\nabla F(\mathbf{x}_{t-\tau})$ are negatively correlated, and $\{\nabla F(\mathbf{x}_t)\}$ may contain high-frequency signals.

Before we present the theorem, let us define the normalized SNR as

$$\underline{\mathbf{SNR}} = \frac{\sum_{t=0}^{T-1} \sum_{\tau=0}^t c_\tau \kappa_\tau}{\sum_{t=0}^{T-1} \sum_{\tau=0}^t \kappa_\tau^2}, \tag{5}$$

and define the expanded coefficients $\kappa_\tau$ as

$$\kappa_\tau = \sum_{\tau_2=0}^{\min\{n_b, \tau\}} b_{\tau_2} \sum_{\tau_1=1}^{n_a} z_{a,\tau_1} (p_{a,\tau_1})^{\tau-\tau_2}, \quad \text{s.t.} \quad \sum_{\tau=1}^{n_a} \frac{z_{a,\tau}}{1 - p_{a,\tau}x} = \frac{1}{1 + \sum_{\tau=1}^{n_a} a_\tau x^\tau}, \tag{6}$$

which satisfies $m_t = -\sum_{\tau=1}^{n_a} a_\tau m_{t-\tau} + \sum_{\tau=0}^{n_b} b_\tau g_{t-\tau} = \sum_{\tau=0}^t \kappa_\tau g_{t-\tau}$. Note that $p_{a,\tau}$ might be complex, but $\kappa_\tau$ are guaranteed to be real. With A4, we have the following convergence result for Algorithm 2.

**Theorem 2 (Convergence)** *Assume the problem satisfies A1-A4. By choosing $C \geq G$, $\eta \leq \min\{\frac{2c_{-\tau}}{L\kappa_\tau}\}$, and running Algorithm 2 for $T$ iterations, the algorithm satisfies:*

$$\mathbb{E}_{t \sim P(t)} \|\nabla F(\mathbf{x}_t)\|^2 \leq \frac{F(\mathbf{x}_0) - F^\star}{\eta S_T} + \frac{\eta L}{2\underline{\mathbf{SNR}}} \left(d\sigma_{DP}^2 + \frac{\sigma_{SGD}^2}{B}\right), \tag{7}$$

*where we define $S_T = \sum_{t=0}^{T-1} \sum_{\tau=0}^t c_\tau \kappa_\tau = \mathcal{O}(T)$; the expectation is taken over $t = 0, \ldots, T-1$, such that $P(t) = \frac{\sum_{\tau=0}^t c_\tau \kappa_\tau}{S_T}$.*

The proof of Theorem 2 is given in Appendix B. Compared with vanilla DPSGD (Abadi et al., 2016), by adopting DOPPLER the noise is scaled by a factor of $\frac{1}{2\underline{\mathbf{SNR}}}$. Thus, as long as $\underline{\mathbf{SNR}} > \frac{1}{2}$, the noise is reduced. Next, we will use the above result to obtain privacy-utility tradeoff.

## 5.2 Privacy guarantee

Our low-pass filter is post-processing on the privatized gradient. Since DP is immune to post-processing Dwork and Roth (2014), Algorithm 2 provides the same DP guarantee as DPSGD, satisfying Theorem 1. By directly combining Theorem 1 and Theorem 2, we can obtain the following privacy-utility trade-off for Algorithm 2.

**Theorem 3 (Privacy-utility trade-off)** *Assume the problem satisfies A1-A4. By choosing $C = G$, $\sigma_{DP}^2 = \frac{C^2 T \ln(1/\delta)}{N^2 \epsilon^2}$, $\eta \leq \min\{\frac{2c_{-\tau}}{L\kappa_\tau}\}$, and running Algorithm 2 for $T = \mathcal{O}\left(\frac{N\epsilon\sqrt{\underline{\mathbf{SNR}}(F(\mathbf{x}_0) - F^\star)}}{C\sqrt{dL\ln(1/\delta)}}\right)$ iterations, the algorithm satisfies $(\epsilon, \delta)$-DP and the expected gradient satisfies:*

$$\mathbb{E}_{t \sim P(t)} \|\nabla F(\mathbf{x}_t)\|^2 = \mathcal{O}\left(\frac{C\sqrt{dL(F(\mathbf{x}_0) - F^\star)\ln(1/\delta)}}{\sqrt{\underline{\mathbf{SNR}}}N\epsilon}\right),$$

*where $P(t) = \frac{\sum_{\tau=0}^t c_\tau \kappa_\tau}{\sum_{t=0}^{T-1} \sum_{\tau=0}^t c_\tau \kappa_\tau}$ and $\kappa, \underline{\mathbf{SNR}}$ are defined in (6), (5), respectively.*

Theorem 3 implies that DPSGD with DOPPLER shares the same convergence rate $\mathcal{O}\left(\frac{C\sqrt{d\ln(1/\delta)}}{N\epsilon}\right)$ as the vanilla DPSGD (Abadi et al., 2016). However, by using the low-pass filter, the performance of DPSGD improves by a *constant factor* $\frac{1}{\sqrt{\underline{\mathbf{SNR}}}}$, which is discussed next.

## 5.3 Impact of the low-pass filter

Here, we provide $\underline{\textbf{SNR}}$ value for some choices of the filter coefficients and discuss how to design low-pass filters.

- For SGD (no filter), we have $\kappa_0 = 1$, and $\kappa_\tau = 0, \ \forall \tau > 0$. Then, the normalized SNR is $\underline{\textbf{SNR}} = \frac{1}{2}$. This recovers the convergence result for DPSGD in Abadi et al. (2016).

- Momentum-SGD Cutkosky and Mehta (2020) is a special case of the low-pass filter, with filter coefficients: $a_1 = -0.9, b_0 = 0.1$. and $\kappa_\tau = 0.1 \times 0.9^\tau$. Then, the normalized SNR is $\underline{\textbf{SNR}} \geq 1.9 \times \left( \frac{1}{2} + \sum_{\tau=0}^{t-1} 0.9^\tau c_\tau \right)$, which is larger than vanilla DPSGD. This indicates that DPSGD with momentum can reduce the impact of DP noise compared with DPSGD w/o momentum.

- We can further improve the SNR by optimizing the filter coefficients under a fixed order:

$$\max_{\{a_\tau\}, \{b_\tau\}} \quad \underline{\textbf{SNR}}, \quad \text{s.t.} \quad \sum_{\tau=0}^{n_b} b_\tau - \sum_{\tau=1}^{n_a} a_\tau = 1. \tag{8}$$

From (8), we observe that the pattern of the auto-correlation coefficients $c_\tau$ determines the choice of the filter coefficients. When $\kappa_\tau \propto c_\tau$, $\underline{\textbf{SNR}}$ is maximized. However, in general, finding the *optimal* filter coefficients by optimizing (8) before training is difficult, as $\{c_\tau\}$ is determined by the problem and the DP optimizer's updates and can be time-varying.

**Optimal FIR filter:** When the filter takes the form of a finite impulse response (FIR) (i.e., $a_n = 0$), we can estimate $\{c_\tau\}$ and optimize $b_\tau$'s according to (8) during training. To estimate $c_\tau$, we have:

$$c_\tau \overset{A4}{\leq} \left( \langle \nabla F(\mathbf{x}_t), \nabla F(\mathbf{x}_{t-\tau}) \rangle - c_{-\tau} \| \nabla F(\mathbf{x}_{t-\tau}) \|^2 \right) / \| \nabla F(\mathbf{x}_t) \|^2$$

$$\overset{A2}{\leq} \mathbb{E} \left[ \langle \mathbf{g}_t, \mathbf{g}_{t-\tau} \rangle - c_{-\tau} \left( \| \mathbf{g}_{t-\tau} \|^2 - d\sigma_{\text{DP}}^2 - \sigma_{\text{SGD}}^2/B \right) \right] / \mathbb{E} \left[ \| \mathbf{g}_t \|^2 - d\sigma_{\text{DP}}^2 - \sigma_{\text{SGD}}^2/B \right]$$

$$\approx \mathbb{E} \left[ \langle \mathbf{g}_t, \mathbf{g}_{t-\tau} \rangle - \frac{1}{2} \max\{ \| \mathbf{g}_{t-\tau} \|^2 - d\sigma_{\text{DP}}^2, 0 \} \right] / \mathbb{E} \left[ \max\{ \| \mathbf{g}_t \|^2 - d\sigma_{\text{DP}}^2, \epsilon_1 \}, \right]$$

where in the last approximation we set $c_{-\tau} = \frac{1}{2}$ and assume $d\sigma_{\text{DP}}^2$ dominates $\sigma_{\text{SGD}}^2/B$; the $\max$ are taken as $\|\cdot\|^2 \geq 0$ and we choose $\epsilon_1 = 10^{-3}$ as a small positive number for numerical stability. After obtaining $c_\tau'$s, $b_\tau'$s have a closed-form solution $b_\tau = \frac{c_\tau}{\sum_{\tau=0}^{b_n} c_\tau}$. This estimation only relies on the stored privatized gradients $\{\mathbf{g}_{t-\tau}\}_{\tau=0}^{n_b}$, so it does not spend an extra privacy budget or memory. Therefore, it can also be implemented along with DOPPLER, as an adaptive approach to adjust the filter coefficients for an optimal performance.

## 6 Numerical experiments

In this section, we investigate how the low-pass filter affects the performance of various DP optimizers on different datasets, privacy budgets, and models. Due to the page limitation, detailed implementation and extra numerical results are given in Appendix C. The code for the experiments is available at `https://anonymous.4open.science/r/Low-pass-SGD-C7A1`.

### 6.1 Experiment Settings

**Dataset:** We conduct experiments on computer vision datasets (MNIST, CIFAR-10, and CIFAR-100 (Krizhevsky et al., 2009)) and natural language processing datasets, GLUE (Wang et al., 2018).

**Model:** We conduct experiments on various models, including the 5-layer CNN described in De et al. (2022), the modified ResNet in Kolesnikov et al. (2019), EfficientNet with group normalization (Tan and Le, 2019), and ViT (Dosovitskiy et al., 2020) for the CV tasks and RoBERTa-base (Liu et al., 2019) for the GLUE dataset. If not specified, the models are initialized with random weights *without pretraining*.

**Algorithm:** We compared the impact of DOPPLER on several base algorithms, including the DP version of SGD, Adam, and GaLore. The updates of the algorithms are given in Algorithm 2 and Algorithm 3 in Appendix C.2. We use **LP-** to denote the DP optimizer with DOPPLER.

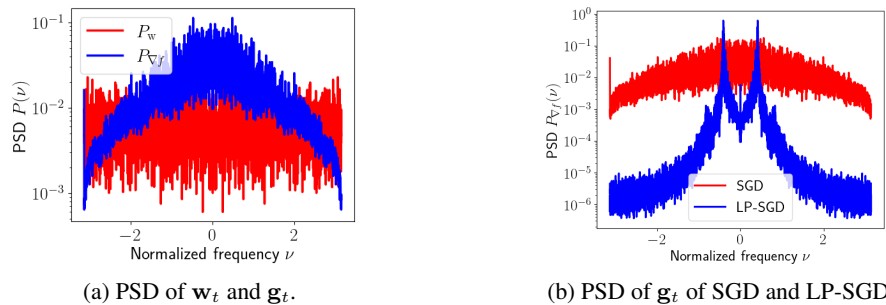

(a) PSD of $\mathbf{w}_t$ and $\mathbf{g}_t$.      (b) PSD of $\mathbf{g}_t$ of SGD and LP-SGD.

Figure 2: The recorded PSD of Gaussian noise $\{\mathbf{w}_t\}$, and the stochastic gradients of SGD and LP-SGD of ResNet-50 training on CIFAR-10 dataset.

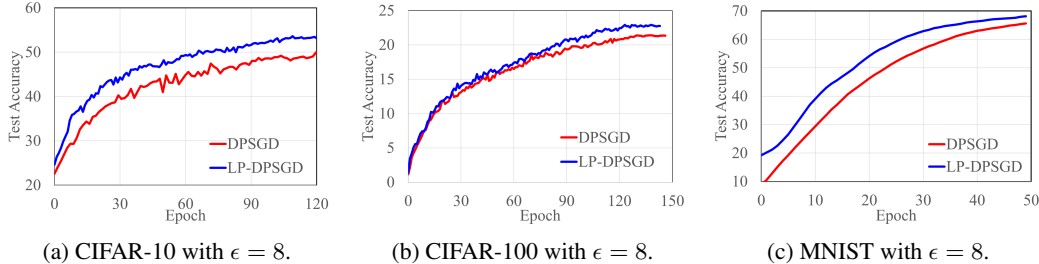

(a) CIFAR-10 with $\epsilon = 8$.    (b) CIFAR-100 with $\epsilon = 8$.    (c) MNIST with $\epsilon = 8$.

Figure 3: Comparision between DPSGD and LP-DPSGD for pre-training on different datasets.

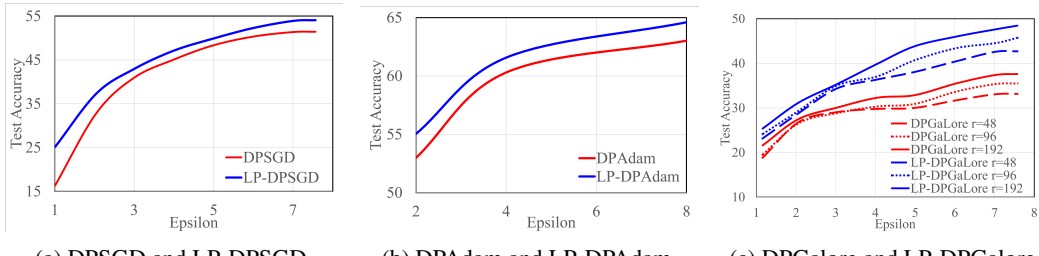

(a) DPSGD and LP-DPSGD.    (b) DPAdam and LP-DPAdam.    (c) DPGalore and LP-DPGalore.

Figure 4: Comparision between DP optimizers w and w/o low-pass filters for pre-training with different $\epsilon$'s on CIFAR-10 dataset.

**Hyper-parameters choices:** The choices of the filter coefficients $a_i, b_i$ are empirical; specific choices used in the experiments are listed in Table 2 in Appendix C.1.

The learning rate, batch size, and number of training epochs are tuned for best testing accuracy using grid search. Detailed hyper-parameters and search grids are given in Appendix C.1. For all experiments, we fix the privacy parameter $\delta = 1/N^{1.1}$ to obtain a reasonable privacy notion.

## 6.2 Numerical results

**PSD of the stochastic gradient:** First, we record the stochastic gradients of SGD and SGD with DOPPLER training ResNet-50 for 40 epochs ($T = 4000$ steps) on the CIFAR-10 dataset. Then, we compute the PSD of the recorded stochastic gradients. The results are given in Figure 2. We can observe that the recorded PSD of $\mathbf{w}_t$ is filling all frequencies, and $\mathbf{g}_t$ is a low-frequency signal. After applying the low-pass filter, the PSD of the filtered gradient lies in the low-frequency domain, and the high-frequency signals (and noise) are suppressed.

**Results for different datasets.** The comparisons between LP-DPSGD and DPSGD on different datasets are given in Figure 3. We can observe that LP-DPSGD outperforms DPSGD on MNIST, CIFAR-10, and CIFAR-100 datasets under the same privacy budget $\epsilon = 8$. The results on the GLUE dataset is in Appendix C.3.

**Results for different algorithms:** The comparisons between DP optimizers, including DPSGD, DPAdamBC (Tang et al., 2024), and DPGaLore (an extension of GaLore (Zhao et al., 2024)), with and without DOPPLERis shown in Figure 4. We can observe that all DP optimizers with DOPPLERoutperform the baseline under different levels of privacy budget $\epsilon$'s.

# 7 Conclusion and discussion

In this paper, we introduce a signal processing perspective to understand and analyze DP optimizers. By identifying the difference between the gradient and noise signal in the frequency domain, we propose DOPPLER, a low-pass filter approach, to filter out the DP noise and improve the signal-to-noise ratio of the privatized gradient. Our proposed filtering method is compatible with existing DP optimizers, and extensive experiments have shown that the low-pass filter could improve DP optimizers' performance in the case when the DP noise is large, e.g., in the pertaining stage and for training large models.

**Limitations:** Designing higher-order filters requires hyper-parameter tuning or prior knowledge of the gradients' auto-correlation pattern; implementing a high-order low-pass filter is memory inefficient (requires storing $n_a + n_b$ optimization states for each trainable parameter), which eliminates the usage of the proposed method when optimizing very large-scale foundation models with limited memory resource.

## Acknowledgements

This work is supported by a gift from the USC-Meta Center for Research and Education in AI, and a gift from Google. Mingyi Hong is supported partially by NSF under the grants EPCN-2311007 and CCF-1910385.

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

# A    Additional Background

In this section, we provide the details of the frequency-domain analysis and low-pass filter approach. First, we discuss the basic concepts in signal processing. Then, we discuss the filter method in signal processing.

## A.1    Frequency domain analysis

In signal processing, frequency domain analysis is used to analyze the periodical or long-term behavior of a (time series) signal/data. In the frequency domain analysis, we use the frequency $\nu$ as the indices of the signal, e.g., $\{X(\nu)\}, X(\nu) \in \mathbb{C}$, where each term $X(\nu)$ records the amplitude and phase of the sine wave of frequency $\nu$ that composes the signal; in contrast, in the time domain, we use time $t$ as the indices of a signal, e.g., $\{x_t\}$, where each term $x_t$ records the value of the signal at a given time $t$. In this paper, we treat each coordinate $i \in [1, \ldots, d]$ of the privatized gradient over the iterates as an individual signal, i.e. $\{g_1[i], g_2[i], \ldots, g_T[i]\}$. Thus, the gradient over iterates gives us $d$ one-dimensional signals, and we can look at their frequency domain representation of each signal.

**Benefit of frequency-domain analysis:**

- Certain properties of a signal can be hard to observe/characterize in the time domain. For example, a long-time correlation or a cyclic behavior of the signal is not easy to directly observe in the time domain. By converting the signal to the frequency domain, such properties can easily be captured and analyzed. For example, the signal $x_t = \sin(t)$ has nonzero entries in almost all times. However, the frequency domain representation of this signal has only one entry that is non-zero, i.e., $X(1) = 1$ and all other entries are zero, i.e., $X(\nu) = 0, \forall \nu \neq 1$. This means $x_t$ has only one periodic signal in it.

- Certain mathematical analyses can be significantly simplified in the frequency domain. For example, linear differential equations in the time domain are converted to algebraic equations in the frequency domain; filters as convolutions in the time domain are converted to point-wise multiplication in the frequency domain. These properties greatly simplify the analysis of the dynamics of the signals and filters (Oppenheim et al., 1996).

**Transform from time to frequency domain:** To obtain a frequency domain representation of a discrete signal, one can apply the Discrete Fourier transform (DFT) ($\mathcal{F}\{x_t\} : X(\nu) = \sum_{t=0}^{T-1} x(t)e^{\frac{-2\pi it}{T}\nu}$)) to the signal. By directly applying DFT to a signal and obtaining $\{X(\nu)\}$, one can identify how the signal is composed of sin waves of different frequencies $\nu$ with their amplitudes and phases. In the paper, we apply DFT to the auto-correlation of a signal and obtain its power spectrum density (PSD). The PSD of a signal shows the distribution of the power of a signal on different frequencies. For example, the PSD of $x(t) = \sin(t)$ is $P(\nu) = 1/2$ for $\nu = \pm\frac{1}{2\pi}$ and 0 elsewhere.

## A.2    Low-pass filter

**Frequency filter:** A frequency filter is a transformation of a signal that only allows certain frequencies to pass and blocks/attenuates the remaining frequencies. For example, for a signal $x(t) = \sin(t) + \sin(10t)$, we can apply an (ideal) low-pass filter $F(\nu) = 1$ when $|\nu| \leq \frac{1}{2\pi}$ and 0 otherwise. Then, after applying the filter, $F * x(t) = \sin(t)$, the output signal only keeps the low-frequency signal.

In this work, we use (time-invariant) linear filters for DP noise reduction. A linear filter attenuates certain frequencies by using a linear combination of the input signal. Considering $g_t$ as the time signal, the general form of a linear filter on $g_t$ is

$$m_t = \sum_{\tau=0}^{t} \kappa_\tau g_{t-\tau} = -\sum_{\tau=1}^{n_a} a_\tau m_{t-\tau} + \sum_{\tau=0}^{n_b} b_\tau g_{t-\tau},$$

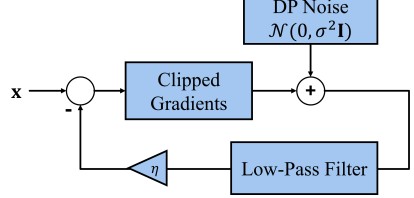

Figure 5: Illustration of the low-pass filter.

where $\kappa_\tau$ are the filter coefficients. The second formula is a recursive way of writing the filter.

**Filter design:** The property of the filter depends on the choice of the filter coefficients. Designing a filter consists of the following steps:

- **Decide filter order/tab** $n_a, n_b$. Larger $n_a, n_b$ give the filter more flexibility and better possible performance, at a cost of more memory consumption. In our experiment, we tested on 0th-3rd order filters, i.e., $\max\{n_a, n_b\} \le 3$.

- **Decide filter coefficients** $\{a_\tau\}, \{b_\tau\}$. Filter design can, in general, be a complex procedure, and it involves deciding on trade-offs among different properties of the filter (Winder, 2002). Two standard constraints on the filter coefficients are: a) $-\sum a_\tau + \sum b_\tau = 1$, to ensure the filter has unit gain, i.e., the mean of the signal remains unchanged; and b) the solutions $x$ to $1 + \sum a_\tau x^\tau = 0$ satisfies $|x| < 1$, to ensure the filter is stable, i.e., $\sum |\kappa_t| < \infty$. In the paper, we directly follow the design of the Chebyshev filter and Butterworth filter and tune their cut-off frequency (and ripple) to achieve the best performance and maintain these properties.

### A.3 Additional related work

**DP optimization with correlated noise:** DP optimization with correlated noise have been investigated in Kairouz et al. (2021); Denisov et al. (2022); Choquette-Choo et al.. These works treat the DPSGD update as releasing a *weighted prefix sum* with DP noise, i.e., $A(G_{0:t} + W_{0:t})$, where $A$ is the prefix sum matrix (a lower-triangular all-one matrix) and $W_{0:t}$ is the i.i.d. DP noise. Kairouz et al. (2021); Denisov et al. (2022) apply certain decomposition $A = BC$ and change the update to $B(CG_{0:t} + W_{0:t}) = AG_{0:t} + BW_{0:t}$, and Choquette-Choo et al. provides a theoretical justification that when $B$ is a high-pass filter, and $g_t$ are correlated, the algorithm outperforms original DPSGD. In contrast, our method can be written as $AM(G_{0:t} + W_{0:t})$, where $M$ is a low-pass filter.

- The correlated noise methods and our proposed method can all be viewed as processing the signal in the frequency domain to "separate" the noise and the gradient.

- The existing correlated noise methods 1) pre-process the gradient/noise to separate the gradient and the DP noise in the frequency domain, and therefore require careful design of the matrices $B, C$ for each problem and optimizer, 2) require extra memory ($O(d \log(t))$ to $O(dt)$), which is unrealistic for large scale training, and 3) only work for SGD update, since Adam cannot be written as such a prefix sum of privatized gradient. In contrast, our method 1) post-processes the noisy signal to extract the gradient from the noise from the frequency domain, 2) only requires $O(d)$ extra memory, which is independent of $t$, and 3) is compatible with any first-order optimizer since it just post-processes the gradient.

## B   Missing proof details in the main paper

In this section, we provide the proof for Theorem 2. First, by A1, we have

$$F(\mathbf{x}_{t+1}) - F(\mathbf{x}_t) \le \langle \nabla F(\mathbf{x}_t), \mathbf{x}_{t+1} - \mathbf{x}_t \rangle + \frac{L}{2} \mathbb{E}\left[\|\mathbf{x}_{t+1} - \mathbf{x}_t\|^2\right]$$
$$= -\eta \langle \nabla F(\mathbf{x}_t), \mathbf{m}_t \rangle + \frac{L\eta^2}{2} \|\mathbf{m}_t\|^2. \tag{9}$$

We can expand $\mathbf{m}_t$ as follows:

$$\mathbf{m}_t = -\sum_{\tau=1}^{n_a} a_\tau \mathbf{m}_{t-\tau} + \sum_{\tau=0}^{n_b} b_\tau \mathbf{g}_{t-\tau}$$

$$\mathbf{m}_t + \sum_{\tau=1}^{n_a} a_\tau \mathbf{m}_{t-\tau} = \sum_{\tau=0}^{n_b} b_\tau \mathbf{g}_{t-\tau}$$

$$(1 + \sum_{\tau=1}^{n_a} a_\tau z^{-\tau})\mathbf{M}(z) \overset{(a)}{=} (\sum_{\tau=0}^{n_b} b_\tau z^{-\tau})\mathbf{G}(z)$$

$$\mathbf{M}(z) \overset{(b)}{=} \frac{1}{1 + \sum_{\tau=1}^{n_a} a_\tau z^{-\tau}} (\sum_{\tau=0}^{n_b} b_\tau z^{-\tau})\mathbf{G}(z)$$

$$\mathbf{M}(z) \overset{(c)}{=} \sum_{\tau=1}^{n_a} \frac{z_{a,\tau}}{1 - p_{a,\tau} z^{-1}} (\sum_{\tau=0}^{n_b} b_\tau z^{-\tau}) \mathbf{G}(z)$$

$$\mathbf{M}(z) \overset{(d)}{=} \sum_{\tau_1=1}^{n_a} z_{a,\tau_1} \sum_{\tau_0=0}^{\infty} (p_{a,\tau_1} z^{-1})^{-\tau_0} (\sum_{\tau_2=0}^{n_b} b_{\tau_2} z^{-\tau_2}) \mathbf{G}(z)$$

$$\mathbf{M}(z) \overset{(e)}{=} \sum_{\tau_2=0}^{n_b} b_{\tau_2} \sum_{\tau_1=1}^{n_a} z_{a,\tau_1} \sum_{\tau_0=0}^{\infty} (p_{a,\tau_1})^{\tau_0} z^{-\tau_0 - \tau_2} \mathbf{G}(z)$$

$$\mathbf{m}_t \overset{(f)}{=} \sum_{\tau_2=0}^{n_b} b_{\tau_2} \sum_{\tau_1=1}^{n_a} z_{a,\tau_1} p_{a,\tau_1}^{\tau_0} \mathbf{g}_{t-\tau_0-\tau_2}$$

$$= \sum_{\tau=0}^{t} \kappa_\tau \mathbf{g}_{t-\tau},$$

where in $(a)$ we applies the $z$-transform $\mathcal{Z}\{\cdot\}$ to both side of the sequence and use the property that $\mathcal{Z}\{x_{t-k}\} = z^{-k}\mathbf{X}(z)$ (Oppenheim et al., 1996); $(b)$ divides both sides by $1 + \sum_{\tau=1}^{n_a} a_\tau z^{-\tau}$; in $(c)$ we define $\{z_{a,\tau}\}, \{p_{a,\tau}\}$ such that $\sum_{\tau=1}^{n_a} \frac{z_{a,\tau}}{1 - p_{a,\tau} z^{-1}} = \frac{1}{1 + \sum_{\tau=1}^{n_a} a_\tau z^{-\tau}}$; $(d)$ expands $\frac{1}{1-p} = \sum_{t=0}^{\infty} p^t$; in $(e)$ we rearrange there terms; in $(f)$ we apply the inverse $z$-transform and notice that $\mathbf{g}_{<0} = 0$; and in the last equation we define $\kappa_\tau = \sum_{\tau_2=0}^{\min\{n_b,\tau\}} b_{\tau_2} \sum_{\tau_1=1}^{n_a} z_{a,\tau_1} (p_{a,\tau_1})^{\tau-\tau_2}$. Plug the expansion of $\mathbf{m}_t$ back to (9), we have

$$\mathbb{E}[F(\mathbf{x}_{t+1}) - F(\mathbf{x}_t)] \leq -\eta \left\langle \nabla F(\mathbf{x}_t), \mathbb{E}[\sum_{\tau=0}^{t} \kappa_\tau \mathbf{g}_{t-\tau}] \right\rangle + \frac{L\eta^2}{2} \mathbb{E}\left[ \left\| \sum_{\tau=0}^{t} \kappa_\tau \mathbf{g}_{t-\tau} \right\|^2 \right]$$

$$\overset{(a)}{\leq} -\eta \sum_{\tau=0}^{t} \kappa_\tau \langle \nabla F(\mathbf{x}_t), \nabla F(\mathbf{x}_{t-\tau}) \rangle + \frac{L\eta^2}{2} \left[ \left\| \sum_{\tau=0}^{t} \kappa_\tau \mathbf{g}_{t-\tau} \right\|^2 \right]$$

$$\overset{(b)}{\leq} -\eta \sum_{\tau=0}^{t} \kappa_\tau \langle \nabla F(\mathbf{x}_t), \nabla F(\mathbf{x}_{t-\tau}) \rangle$$

$$+ \frac{L\eta^2}{2} \left( \left\| \sum_{\tau=0}^{t} \kappa_\tau \nabla F(\mathbf{x}_{t-\tau}) \right\|^2 + \sum_{\tau=0}^{t} \kappa_\tau^2 \left( d\sigma_{\text{DP}}^2 + \frac{\sigma_{\text{SGD}}^2}{B} \right) \right)$$

$$\overset{(c)}{\leq} -\eta \sum_{\tau=0}^{t} \kappa_\tau c_\tau \|\nabla F(\mathbf{x}_t)\|^2 - \eta \sum_{\tau=0}^{t} \kappa_\tau c_{-\tau} \|\nabla F(\mathbf{x}_{t-\tau})\|^2$$

$$+ \frac{L\eta^2}{2} \left( \sum_{\tau=0}^{t} \kappa_\tau \|\nabla F(\mathbf{x}_{t-\tau})\|^2 + \sum_{\tau=0}^{t} \kappa_\tau^2 \left( d\sigma_{\text{DP}}^2 + \frac{\sigma_{\text{SGD}}^2}{B} \right) \right)$$

$$\overset{(d)}{\leq} -\eta \sum_{\tau=0}^{t} \kappa_\tau c_\tau \|\nabla F(\mathbf{x}_t)\|^2 + \frac{L\eta^2}{2} + \sum_{\tau=0}^{t} \kappa_\tau^2 \left( d\sigma_{\text{DP}}^2 + \frac{\sigma_{\text{SGD}}^2}{B} \right) \tag{10}$$

where in $(a)$ we apply A3 and set $C \geq G$, so that $\mathbb{E}[\mathbf{g}_t] = \nabla F(\mathbf{x}_t)$; $(b)$ applies A2 to the last term, and use the fact that $\mathbf{w}_t \sim \mathcal{N}(0, \sigma_{\text{DP}}^2 \cdot I_d)$, and the noise are independent; $(c)$ applies A4 to the first term and uses Jensen's inequality to the second term with $\|\cdot\|^2$ being convex; in $(d)$ we set $\eta \leq \min_\tau \{\frac{2c_{-\tau}}{L\kappa_\tau}\}$. Clearly, we have $\sum_{\tau=0}^{t} \kappa_\tau \leq \frac{\sum_{\tau=0}^{n_b} b_\tau}{1 - \sum_{\tau=1}^{n_a} a_\tau}$. Averaging over $t = 0, \ldots, T-1$, and deciding both side by $\eta$, then the theorem is proved.

## C   Missing experiment details in the main paper

In this section, we provide the missing details for the experiments in the main paper and additional experiments.

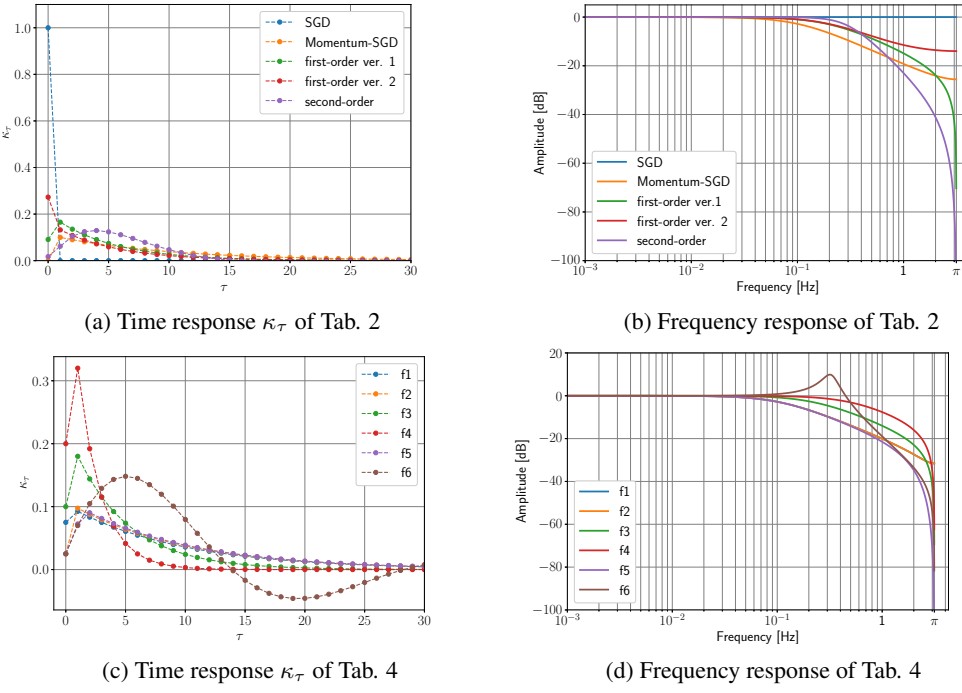

(a) Time response $\kappa_\tau$ of Tab. 2

(b) Frequency response of Tab. 2

(c) Time response $\kappa_\tau$ of Tab. 4

(d) Frequency response of Tab. 4

Figure 6: The time and frequency response of the filters used in the paper.

**Experiments compute resources:** Each experiment is conducted on an EPYC-7513 CPU with one NVIDIA A100 (80 GB) GPU. The runtime ranges from 1 hour to 48 hours. The codes for the optimizers and the job scripts are given in the supplementary.

## C.1 Hyper-parameter choice

In this section, we provide the hyper-parameter choices and search grids for different experiment settings. The search grids for the experiments are given in Table 1.

Table 1: Search grids for each hyper-parameter.

| Hyper-parameter | Search grid |
|---|---|
| Epoch | $\{50, 80, 120, 150\}$ |
| CIFAR10/CIFAR100 batch size | $\{500, 1000, 2000, 5000\}$ |
| GLUE batch size | $\{1024, 2048, 4096\}$ |
| SGD stepsize | $\{0.8, 0.5, 0.4, 0.2, 0.1\}$ |
| Adam/GaLore stepsize | $\{3e-3, 1e-3, 3e-4, 1e-4, 3e-5\}$ |

The filter coefficients used in the experiments are given in Table 2.

Table 2: Possible choice of the coefficients for the filter of different orders.

| Filter | $b_\tau$ | $a_\tau$ |
|---|---|---|
| SGD | 1 | N/A |
| Momentum-SGD | 0.1 | $-0.9$ |
| $1^{\text{st}}$-order ver.1 | $\{1, 1\}/11$ | $-9/11$ |
| $1^{\text{st}}$-order ver.2 | $\{3, -1\}/11$ | $-9/11$ |
| $2^{\text{nd}}$-order | $\{1, 2, 1\}/58$ | $\{-92, 38\}/58$ |

## C.2 Algorithm variants

We provide the update rules for different DP optimizer variants used in the main paper in Algorithm 3. Different components of the optimizers are highlighted with different colors. the blue lines are additional steps for the Adam update, and brown lines are the components of the GaLore (Zhao et al., 2024) optimizer.

**Algorithm 3** DP-GaLore with DOPPLER

1: **Input:** $\mathbf{x}_0, \mathcal{D}, \eta, C, \sigma_{\text{DP}}, \{a_\tau\}_{\tau=1}^{n_a}, \{b_\tau\}_{\tau=0}^{n_b}, \beta, \epsilon_{\text{Adam}}, I, r$
2: **Initialize:** $\{\mathbf{m}_{-\tau}\}_{\tau=1}^{n_a} = 0, \{\mathbf{g}_{-\tau}\}_{\tau=1}^{n_b} = 0, \{c_{a,-\tau}\}_{\tau=1}^{a_n}, \{c_{b,-\tau}\}_{\tau=0}^{b_n}, \mathbf{v}_{-1} = 0$
3: **for** $t = 0, \ldots, T-1$ **do**
4:      Randomly draw minibatch $\mathcal{B}_t$ from $\mathcal{D}$
5:      $\mathbf{g}_t = \frac{1}{|\mathcal{B}_t|} \sum_{\xi \in \mathcal{B}_t} \text{clip}\left(\nabla f(\mathbf{x}; \xi), C\right) + \mathbf{w}_t$          *# Compute private gradient*
              where $\mathbf{w}_t \sim \mathcal{N}(0, \sigma_{\text{DP}}^2 \cdot \mathbf{I}_d)$
6:      **if** Use GaLore **then**
7:          **if** $t \mod I \equiv 0$ **then**
8:              $\mathbf{P} = \text{FindProjector}(\mathbf{g}_t, r)$
9:          **end if**
10:        $\tilde{\mathbf{g}}_t = \mathbf{P}^\top \mathbf{g}_t$          *# Low-dimension projection*
11:      **else**
12:          $\tilde{\mathbf{g}}_t = \mathbf{g}_t$
13:      **end if**
14:      $\mathbf{v}_t = (1-\beta)\mathbf{v}_{t-1} + \beta(\tilde{\mathbf{g}}_t)^2$          *# Compute 2nd-order moment*
15:      $\mathbf{m}_\tau = -\sum_{\tau=1}^{n_a} a_\tau \mathbf{m}_{t-\tau} + \sum_{\tau=0}^{n_b} b_\tau \tilde{\mathbf{g}}_{t-\tau}$          *# Apply filter*
16:      $c_{b,t} = 1, c_{a,t} = -\sum_{\tau=1}^{n_a} a_\tau c_{a,t-\tau} + \sum_{\tau=0}^{n_b} b_\tau c_{b,t-\tau}$          *# Compute bias*
17:      $\hat{\mathbf{m}}_t = \mathbf{m}_t / c_{a,t}$          *# Correct initialization bias*
18:      $\hat{\mathbf{v}}_t = \mathbf{v}_t / (1-\beta^t)$          *# Correct initialization bias*
19:      $\tilde{\mathbf{d}}_t = \hat{\mathbf{m}}_t / \max\{\sqrt{\hat{\mathbf{v}}_t}, \epsilon_{\text{Adam}}\}$
20:      **if** Use GaLore **then**
21:          $\mathbf{d}_t = \mathbf{P}\tilde{\mathbf{d}}_t$          *# Inverse projection to original dimension*
22:      **else**
23:          $\mathbf{d}_t = \tilde{\mathbf{d}}_t$
24:      **end if**
25:      $\mathbf{x}_{t+1} = \mathbf{x}_t - \eta \mathbf{d}_t$          *# Parameter update*
26: **end for**

## C.3    Additional experiments: the GLUE dataset

We also conduct experiments on the GLUE dataset (Wang et al., 2018). We fine-tune a RoBERTa-base model (Liu et al., 2019) with the pretrained weights from `https://huggingface.co/FacebookAI/roberta-base`. We follow the training script provided in Li et al. (2021). The results are shown in Table 3. For comparison, we also include the results from Li et al. (2021). From the result of DPAdamBC and LP-DPAdamBC, we observe a slight accuracy improvement by using DOPPLER. However, in the fine-tuning tasks, we do not see significant improvement by using DOPPLER compared with the result reported in Li et al. (2021).

Table 3: Test accuracy on language tasks with RoBERTa-base, $\epsilon = \{3, 8\}$.

| Method | $\epsilon = 3$ | | | | $\epsilon = 8$ | | | |
|---|---|---|---|---|---|---|---|---|
| | MNLI | QQP | QNLI | SST-2 | MNLI | QQP | QNLI | SST-2 |
| DPAdam (Li et al., 2021) | 82.45 | 85.56 | 87.42 | 91.86 | 83.20 | 86.08 | 87.94 | 92.09 |
| DPAdamBC | 82.39 | 85.29 | 86.44 | 91.10 | 82.48 | 85.94 | 87.06 | 91.25 |
| LP-DPAdamBC | 83.55 | 85.71 | 87.63 | 91.71 | 83.80 | 86.50 | 87.76 | 91.82 |

## C.4    Additional experiments: ablation studies

In this section, we provide additional numerical results for several ablation studies w.r.t. the impact of different components in the experiment.

**Results for different models.** The results for pretraining different models are given in Figure 7. We observe that DOPPLERuniformly improves the DP pretraining performance for different model structures in pre-training.

**Ablation study on filter coefficients.** The results for DPSGD accompanied with different low-pass filter coefficients in Table 2 are given in Figure 8a. We observe that different filters have different impacts on the algorithm's performance. For training on the CIFAR-10 dataset, the first-order filter is sufficient to have a good performance, while different coefficient choices for filters of the same

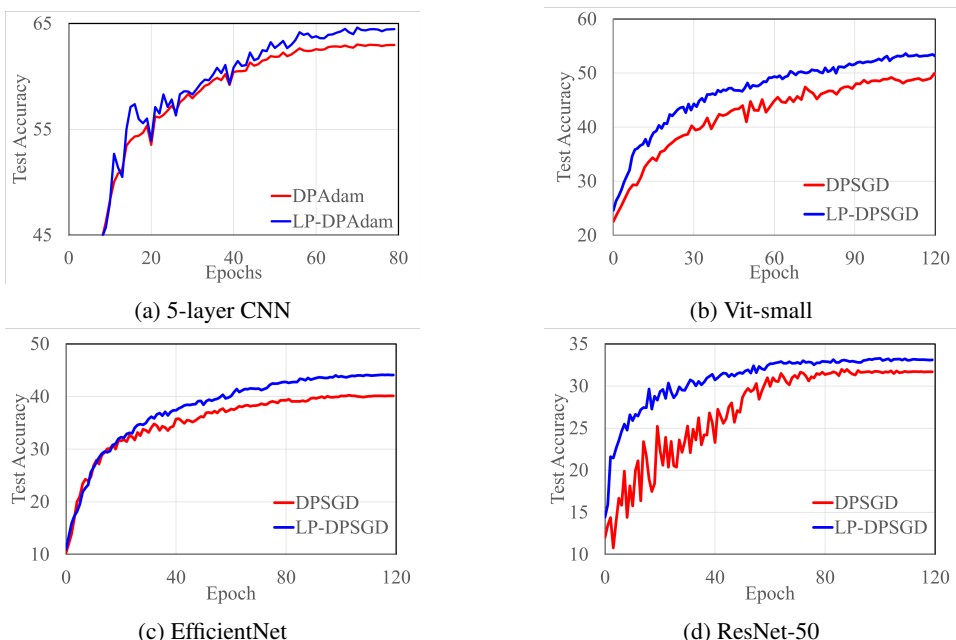

(a) 5-layer CNN

(b) Vit-small

(c) EfficientNet

(d) ResNet-50

Figure 7: Comparision between DPSGD LP-DPSGD for pre-training different models on CIFAR-10 dataset with $\epsilon = 8$.

order also have different performances. The second-order filter provides an over-smoothing to the gradient, leading to slow convergence in the early stage of training. Although the second-order filter does not have a good performance, it could have a better performance when the number of training steps is longer (e.g., when training on the Imagenet dataset (Russakovsky et al., 2015)).

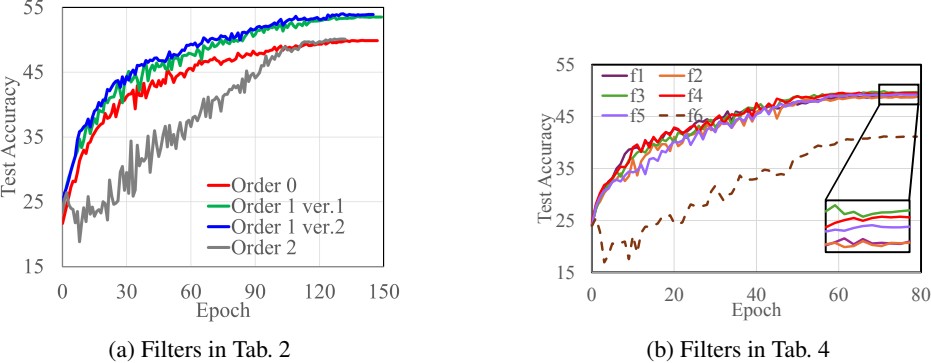

(a) Filters in Tab. 2

(b) Filters in Tab. 4

Figure 8: LP-DPSGD for pre-training on CIFAR-10 with different filter coefficients.

An additional set of first- and second-order filters with different coefficients in Table 4. Coefficient choices f1 and f2 compare different values of $b_\tau$; f3 and f4 compare the impact of the values of $a_\tau$; f5 and f6 compare the impact of filter orders $n_b$ and $n_a$.

Table 4: Possible choice of the coefficients for the filter of different orders.

| Filter | $b_\tau$ | $a_\tau$ |
|---|---|---|
| f1 | $\{0.075, 0.025\}$ | $\{-0.9\}$ |
| f2 | $\{0.025, 0.075\}$ | $\{-0.9\}$ |
| f3 | $\{0.1, 0.1\}$ | $\{-0.8\}$ |
| f4 | $\{0.2, 0.2\}$ | $\{-0.6\}$ |
| f5 | $\{0.025, 0.05, 0.025\}$ | $\{-0.9\}$ |
| f6 | $\{0.025, 0.025\}$ | $\{-1.8, 0.85\}$ |

**Impact of different clipping operations.** The results for LP-DPSGD with different clipping methods are given in Figure 9. We observe automatic clipping (Bu et al., 2024) (Norm) is better than

vanilla clipping described in (Abadi et al., 2016) (Clip); treating all layers as one vector (Flat) is better than clipping each layer separately (Layer).

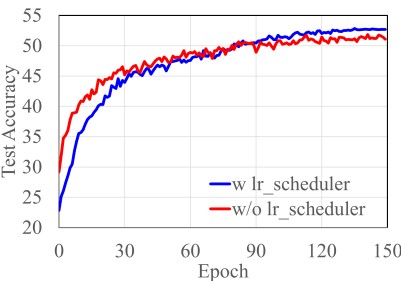

Figure 9: DPLPSGD for pre-training on CIFAR-10 with different clipping strategies.

**Impact of learning rate and scheduler:** We report part of our hyper-parameter search process. First, we report the test accuracy for different learning rates with fixed epochs (150) and $(8, 1/50000^{1.1})$-DP. The results are shown in Figure 10. The optimal learning rate is $10^{-3}$; a larger learning rate speeds up the training in the early stage but hurts the final performance; a smaller learning rate results in slow convergence. In Figure 11, we compare the optimizer with and without a learning rate scheduler. Specifically, we use the Cosine-Annealing with warmup from Loshchilov and Hutter (2017). From the result, we see when the number of epochs is large, the learning rate scheduler improves the training performance, while in the early stage, the scheduler slows down the convergence.

Figure 10: LP-DPAdamBC for pre-training on CIFAR-10 with different learning rates.

Figure 11: LP-DPSGD with and without learning rate scheduler

**Finetuning on CIFAR dataset:** We also provide the experiment result for fine-tuning ViT models on the CIFAR-10 dataset under different privacy budgets. As illustrated in Figure 12, the performance of LP-DPSGD improves by less than 1% for small $\epsilon$'s. For larger $\epsilon$'s, the gap is smaller. DOPPLER works has more improvement when the injected DP noise is large (i.e., $\epsilon$ is small).

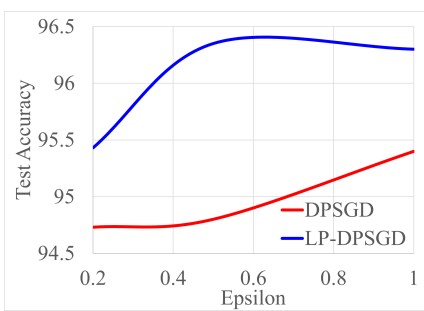

Figure 12: DPSGD and LP-DPSGD for fine-tuning ViT on CIFAR-10 with different $\epsilon$'s.

