# OpenReview forum: "DOPPLER: Differentially Private Optimizers with Low-pass Filter for Privacy Noise Reduction"
_NeurIPS.cc/2024/Conference — NeurIPS 2024 poster_

### Official Review · Reviewer_WMCW · 2024-06-20

**Soundness:** 4
**Presentation:** 3
**Contribution:** 4
**Rating:** 7
**Confidence:** 3

**Summary:**

The paper looks at the sequence of DP-SGD noisy gradients through a signal processing perspective, and argues that the exact gradients are likely a low-frequency signal, while the noise has higher frequencies. As a result, the paper proposes filtering the high-frequencies with a low-pass filter to improve the signal-to-noise ratio. The filter is a post-processing of the noisy gradients, so there is not extra privacy cost. The paper then studies the convergence rate of DP-SGD with the filter under standard assumptions, and finds that the filter can improve the constants of the convergence rate if the hyperparameters of the filter are chosen well. The paper also contains experiments comparing several DP first-order optimiser with and without the filter on several datasets and models. The filtered variant consistently outperforms the unfiltered one.

**Strengths:**

Investigating the behaviour of DP-SGD in the frequency domain is interesting, and to my knowledge novel. The proposed low-pass filter is justified theoretically and consistently improves performance across several models, datasets and optimisers. Since the filter is just a post-processing of the noisy gradients, the privacy analysis does not change, and the filter should be easy to implement with different variants of DP-SGD.

**Weaknesses:**

Many of the signal processing concepts are introduced too briefly, considering the audience, which makes fully understanding the theory difficult. Adding a section to the appendix explaining them clearly would make the theory much easier to understand for people not familiar with signal processing.

The method used to compute the privacy bounds and the subsampling method in the experiments are not mentioned clearly in the paper. "Uniformly draw minibatch" or "randomly draw minibatch" from the algorithm listings is ambiguous, and could mean any of Poisson, with replacement or without replacement subsampling.

Minor points:
- Line 43: 250K steps for LLAMA training sounds like a typo.
- It would be good to mention that there are multiple ways to interpret "neighbourhood" (substitute or add/remove), and that the paper's results do not depend on the precise definition.
- As far as I know, the Gaussian mechanism bound in Definition 2 has only been proven for $\epsilon \leq 1$.
- It is not clear how Algorithm 3 should be read for DPAdam or LP-DPAdam. The update on line 17 uses $\mathbf{d}_t$, but the Adam lines only compute $\tilde{\mathbf{d}}_t$.

**Questions:**

- Do the adaptively selected filter coefficients from Section 5.3 have the same convergence guarantee?
- Did you experiment with adaptively selected filter coefficients from Section 5.3?
- Why does the performance of LP-DPSGD drop with larger epsilons when fine-tuning on CIFAR-10 in Figure 10?

**Limitations:**

The paper clearly mentions the most important limitations of the work.

---

> ### Author Rebuttal · Authors · 2024-08-07
>
> Thank you for reviewing our paper and providing constructive feedback to us. Below, we answered your questions. Please take a look at them and let us know if there are any remaining questions, and we would be more than happy to continue the discussion.
>
> $\quad$
> > Many of the signal processing concepts are introduced too briefly, considering the audience, which makes fully understanding the theory difficult. Adding a section to the appendix explaining them clearly would make the theory much easier to understand for people not familiar with signal processing.
>
> We agree with the reviewer that the discussion on the signal processing would be very helpful to readers with no background in the signal processing domain. We provided a more detailed discussion on the background in our general response to the reviewers. We will expand and add this discussion to the revised manuscript of our paper.
>
> $\quad$
> >The method used to compute the privacy bounds and the subsampling method in the experiments are not mentioned clearly in the paper. "Uniformly draw minibatch" or "randomly draw minibatch" from the algorithm listings is ambiguous, and could mean any of Poisson, with replacement or without replacement subsampling.
>
> Great point! To clarify, we are using the subsampling without replacement strategy. The privacy guarantee is discussed in Balle, et al. 2018. We will clarify this in our revision.
>
> $\quad$
> > Minor points
> >Line 43: 250K steps for LLAMA training sounds like a typo.
>
> According to Touvron et al., 2023, LLAMA-7B/13B are pre-trained with 1T tokens, with a batch size of 4M tokens. Therefore, the total pre-training steps is 1T/4M=250K.
>
> $\quad$
> > It would be good to mention that there are multiple ways to interpret "neighborhood" (substitute or add/remove), and that the paper's results do not depend on the precise definition.
>
> Agreed! We will discuss this point in our revised paper.
>
> $\quad$
> > As far as I know, the Gaussian mechanism bound in Definition 2 has only been proven for $\epsilon\leq 1$.
>
> You are correct that in the original proof in Dwork and Roth, 2014, the guarantee for the Gaussian mechanism is provided only for case $\epsilon \in (0,1)$. While the refined proof for general $\epsilon >0,$ and improved bounds are given by Abadi et al., 2016, and Mironov et al., 2019. We will update the citations in the revised version.
>
> $\quad$
> >It is not clear how Algorithm 3 should be read for DPAdam or LP-DPAdam. The update on line 17 uses $d_t$, but the Adam lines only compute $\tilde{d}_t$.
>
> We apologize for the confusion. If only LP-DPAdam or DPAdam is used, then $\tilde{g}_t = g_t$ in line 9, and $d_t =\tilde{d}_t$ in line 16 in Algorithm 3. We will revise the description of the algorithm.
>
> $\quad$
> > Do the adaptively selected filter coefficients from Section 5.3 have the same convergence guarantee? Did you experiment with adaptively selected filter coefficients from Section 5.3?
>
> This is a great question. Currently, we have neither theoretical results nor numerical experiments for the optimal FIR filter approach proposed in Sec. 5.3. The major focus of this paper is to propose the frequency domain perspective and analysis for DP noise reduction. The filters investigated and analyzed in the paper are *time-invariant*, while the adaptive filter is *time-varying*. We will discuss this briefly in the paper and will leave it as a potential and promising future direction of the paper.
>
> $\quad$
> > Why does the performance of LP-DPSGD drop with larger epsilons when fine-tuning on CIFAR-10 in Figure 10?
>
> Good observation! Notice that the test accuracy drop in such a high accuracy for fine-tuning is quite small (only 0.1%). And the training accuracy in such cases did not drop. Therefore, the performance drop should be attributed to overfitting.
>
> $\quad$
>
> Thank you for reading our rebuttal. We hope the above responses have addressed your concerns. If you have any questions, we would be more than happy to continue discussing them with you.

---

> > ### Comment · Reviewer_WMCW · 2024-08-09
> >
> > Thank you for the response. You have addressed most of my concerns. I especially appreciate the background to signal processing you have in the general response.
> >
> > > While the refined proof for general $\epsilon > 0$ and improved bounds are given by Abadi et al., 2016, and Mironov et al., 2019. We will update the citations in the revised version.
> >
> > Can you point you which of their results you mean? I looked through them quickly and did not find a result which has the same constants as your Definition 2. I'm assuming that by Mironov et al. (2019) you are referring to "Rényi Differential Privacy of the Sampled Gaussian Mechanism", since there is no Mironov et al. (2019) your bibliography.

---

> ### Author Response · Authors · 2024-08-09
> **Further response to Reviewer WMCW**
>
> Thank you for your kind response and for providing additional feedback.
>
> We apologize that we misunderstood your question on $\epsilon <1$ for DP guarantee. Our original response was on Thm.1, which is correct and used in the later proof of our paper.
>
> You are correct on Def. 2 of the Gaussian mechanism that requires $\epsilon <1$. A refined analysis of the Gaussian mechanism is given in Thm. 2, [R1],  which states that $\sigma = \frac{\Delta}{\epsilon}\cdot \frac{\sqrt{2}(a+\sqrt{a^2+\epsilon})}{2},$ with $\mathrm{erfc}(a) - e^\epsilon \mathrm{erfc}(a^2+\epsilon) = 2 \delta.$ This bound works for all $\epsilon>0 , \delta <1.$ We will follow your comment and fix the error in our Def. 2. We would like to point out that we did not use this result in our experiments or proofs in the paper. We just used this definition to introduce the Gaussian mechanism. So we can easily fix this issue and cite proper references. Please also let us know if other references should be included for such a revision.
>
> [R1] Zhao, J., Wang, T., Bai, T., Lam, K. Y., Xu, Z., Shi, S., ... & Yu, H. (2019). Reviewing and improving the Gaussian mechanism for differential privacy. arXiv preprint arXiv:1911.12060.
> Again, thank you for pointing out this issue.

---

### Official Review · Reviewer_vvxz · 2024-06-26

**Soundness:** 3
**Presentation:** 3
**Contribution:** 3
**Rating:** 6
**Confidence:** 4

**Summary:**

This paper augments DP-SGD with a low-pass filter-based postprocessing on the iterates of DP-SGD. This design is based on the intuition that the noise contributes more to the high frequencies, while the gradients (assuming sufficient smoothness of the objective) contribute more to the lower frequencies.

The paper gives a theoretical convergence analysis, showing an improvement corresponding to a signal-to-noise ratio factor depending on the gradient auto-correlation. Importantly, precise knowledge of the gradient autocorrelation is not needed to run the algorithm, but it helps design better filters.

The paper also shows broad empirical improvements from the proposed techniques in the vision setting.

**Strengths:**

- A strength of the proposed method is its simplicity: it gives a simple add-on to the iterates of DP-SGD. It includes momentum as a special case but allows for more sophisticated running averages of the gradients.
- The paper is well-written: the main idea comes across easily (although clarity can be improved; suggestions to follow).
- The theoretical analysis looks sound.
- I think the presented results are significant: The experiments demonstrate a clear win from the proposed approach. They also nicely corroborate the intuition behind the auto-correlation of the gradients vs. noise.
- The first-order filters that work quite well empirically only need a modest additional storage cost (two extra buffers) over DP-SGD.

**Weaknesses:**

While I really like this paper, my opinion is that it is currently lacking in some aspects. Given below are some suggestions to improve the paper along these dimensions.

1. Error bars: DP is a noisy process, so it would be good to see error bars across multiple repetitions of the experiments. This is especially important since the gaps appear to be small. Further, it would be good to see the final accuracies for the vision experiments in a table as it is hard to judge how much the gap between the proposed method and baseline is.

1. Autocorrelation assumption: The general form of Assumption A4 is quite natural given the intuition developed in the preceding sections. However, it is not clear why $c_\tau$ is unconstrained while $c_{-\tau}$ is required to be non-negative. This looks like a trick for the proof to go through but further justification and empirical evidence (potentially in toy problems) are necessary.

1. Clarity: There is much scope for improvement to make the paper more reader-friendly (especially for those unfamiliar with signal processing). Examples:
    - A review section in the appendix recapping the basic ideas of the signal processing tools used (in the appendix)
    - Example coefficients used in typical low pass filters on page 5, and the kinds of weighted averages (the $\kappa_\tau$ coefficients) they would lead to. Also, a few figures to visualize the types of filters that work well with different types of auto-correlations could be helpful.
   - Theorem 2 could be restructured to make it easier to parse. A lot of quantities are used before their definitions.
   - The partial fraction decomposition of eq (7) is not guaranteed to be real. It would be worth clarifying that complex coefficients are allowed. Also, how are the $\kappa_\tau$ coefficients are real?
   - What constraints are necessary on the filter coefficients so that the $\kappa_\tau$ coefficients define a proper weighted average (i.e. non-negative and sum to one)?

1. Parameter tuning: How are the filter coefficients tuned? It is only mentioned that the choices are ``empirical'' in Line 298. Ideally, for momentum SGD, we would tune the momentum and learning rate together; I would expect that tuning the filter coefficients and learning rate together would help.

1. Originality: the proposed approach is a straightforward application of classical ideas from signal processing to private optimization. While this can be viewed as a strength, I would have liked to see a more detailed investigation into the use of low-pass filters. For example:
    - An empirical investigation of the proposed adaptive filter.
    - A detailed empirical study on the choice of the filter coefficients. Only 3 different choices are explored (apart from DP-SGD and momentum), but it would be good to see the effect of changing filter coefficients for a first-order filter.
    - A plot of the performance vs. the order of the filter.
    - Some investigation on observed auto-correlation vs. choice of filter coefficients.

1. Minor comments and Potential typos:
    - what is $t$ in lines 263 and 265?
    - Top of page 13, in equation (c): should $\kappa_\tau \| \nabla F(x_{t-\tau})\|^2$ be $\kappa_\tau^2$ instead?
    - Top of page 13, eq 10: Extra $+$ sign between $\frac{L\eta^2}{2}$ and what follows.

**Questions:**

1. It would be nice to have a more detailed comparison to correlated noise approaches for DP optimization e.g. [KMST+](https://arxiv.org/abs/2103.00039) or [DMRST](https://arxiv.org/abs/2202.08312). For instance, can those approaches be interpreted as some filtering of the noise? Also [CDPG+ (Fig. 1 right)](https://arxiv.org/abs/2310.06771) appear to use a high-pass filter on the noise for DP optimization. This seems to be similar in spirit to the proposed low-pass filter approach and a detailed comparison would be nice.

1. There is no batch size factor in the privacy-utility tradeoff of Theorem 3. How?

1. How does the low-pass filter work on the iterates of DP-SGD instead of the gradients? For instance, it is common to obtain a sequence $x_1, x_2, ...$ from SGD but use the average $\frac{x_1 + \cdots + x_t}{t}$ for inference. I wonder if a sophisticated average from the low-pass filter can help for inference alone (without using it for training).

**Limitations:**

Yes, but a more detailed empirical investigation would be good, as detailed above.

---

> ### Author Rebuttal · Authors · 2024-08-07
>
> Thank you for providing feedback to us. We are glad that you found our paper well-written, the results significant, and the theory meaningful. We are also very excited to hear that you really liked the paper. Below, we respond to your main comments:
>
> > Error bars
>
> The attached PDF provides the error bar for the experiments in Fig.3 (a). We will repeat the other single-run experiments and report all error bars in our revised manuscript.
>
> > Autocorrelation assumption
>
> Great question! The way to look at this assumption is as follows: since there is flexibility in choosing $c_\tau, c_{-\tau}$, we can always choose a small $c_\tau$ so that $c_{-\tau}$ is non-negative, and our theory applies. As you correctly pointed out, the constraint on $c_{-\tau}$ is not restrictive, and is only necessary for simplifying the proof.
>
> > Clarity
>
> Thank you for providing such detailed constructive feedback. We will address these points as discussed next:
>
> > * A review section in the appendix
>
> We will do it as described in our general response to the reviewers.
>
> > * Example coefficients for low-pass filters
>
> We provided the filter coefficients in Table 2 of the  paper. A more detailed discussion on  filter design is given in our general response to the reviewers and will be included in the revised manuscript.
>
> > * Restructuring Thm. 2
>
> Agreed. We will put the definition of quantities ($\underline{SNR}$ and $\kappa$) before the theorem.
>
> > * Construction of $\kappa, p_{a,\tau}$
>
> Indeed, $p_{a,\tau}$, as the solutions to $1 + \sum a_\tau x^\tau = 0$, might not be real. But the resulting weights $\kappa_\tau$ are guaranteed to be real. This is because $\kappa_\tau$ are the weights of the past gradients by recursively expanding  $m_t = -\sum_{\tau=1}^{n_a}a_\tau m_{t-\tau} + \sum_{\tau=0}^{n_b}b_\tau g_{t-\tau} = \sum_{\tau=0}^{t}\kappa_\tau g_{t-\tau}.$ Since $a_\tau, b_\tau$ are real, $\kappa_\tau$'s are also real. We will add this discussion in our revised definition of $\kappa$ before Thm. 2.
>
> > * Constraints on the filter coefficients
>
> Great question! The design of the filter coefficients and the constraints are provided in our general response to the reviewer. In short, our constraints imply that the filter is stable and has a unit gain.
> > Parameter tuning...
>
> The design of the filter coefficients and the constraints are provided in the general response. The other hyper-parameters are tuned using grid search (See App. B.1 Tab. 1 in the original paper) jointly to achieve the optimal performance.
>
> > Originality: the proposed ... For example:
> > * Empirical investigation of the adaptive filter
>
> We agree that adaptive filters are worth further investigation. However, we leave this investigation to future work. This is because this paper focuses on frequency domain analysis, and the investigated filters are *time-invariant*. However, the adaptive  time-varying filters require a different set of analysis tools.
>
> > * On the choice of the filter coefficients
>
> In our PDF response, we further investigated different choices of first-order filter coefficients, in Fig 2. We would expand this and include it in the paper.
>
> > * A plot of the performance vs. the order of the filter
>
> Fig. 6 in the paper investigates the performance of filters of different orders. We observe that a first-order filter performs better than no or second-order filter. A refined version is included in the PDF response, where we compare filters with different combinations of $n_a$ and $n_b$.
> > Minor points
> > * $t$ in lines 263 and 265
>
> From the definition of $\kappa_\tau: m_t = \sum_{\tau=0}^{t}\kappa_\tau g_{t-\tau},$ we see that $\kappa_\tau$ is also a function of $t.$ We apologize for the confusing choice of notations, we will replace $\kappa_\tau$ with $\kappa_{t,\tau}$ in the revised paper.
> > * should $\kappa_\tau |\nabla F(x_{t-\tau})|^2$ be $\kappa_\tau^2$ instead?
>
> We used Jensen's inequlity: $\|\sum\kappa_\tau\nabla F(x_{t-\tau})\|^2 \leq \sum\kappa_\tau\|\nabla F(x_{t-\tau})\|^2,$ with $\sum \kappa_\tau \leq 1, \kappa_\tau \geq 0.$
> > * Top of page 13, eq 10, extra $+$ sign
>
> Agreed, we will fix it.
> > It would be nice to have a more detailed comparison to correlated noise approaches for DP optimization
>
> Thanks for bringing up these related references. Roughly speaking, these methods (KMST+, DMRST, and CDPG+) can be viewed as releasing a *weighted prefix sum* with DP noise, i.e., $A(G_{0:t}+W_{0:t}),$ where $A$ is the prefix sum matrix and $W_{0:t}$ is the i.i.d. DP noise. KMST+ and DMRST apply certain decomposition $A = BC$ and change the update to $B(CG_{0:t}+W_{0:t}) = AG_{0:t}+BW_{0:t},$ and CDPG+ provides a theoretical justification that when $B$ is a high-pass filter, and $g_t$ are correlated, the algorithm outperforms original DPSGD. In contrast, our method can be written as $AM(G_{0:t}+W_{0:t})$, where $M$ is a low-pass filter. We will discuss these methods  in our revised paper. Due to response limitation, the detailed discussion is given in the comment below.
> > No batch size in Thm. 3
>
> From eq(5) in theorem 2, we see that the only place $B$ appears in the proof is in the last term $\sigma^2_{SGD}/B$, which is dominant by the previous term $d\sigma^2_{DP}$. Therefore, the choice of $B$ does not play a critical role in the privacy-utility trade-off in Thm. 3. Similar results can also be found in e.g., Bassily et al., 2014, where $B$ also does not appear in the final bound.
> > LP filter on iterates of DP-SGD?
>
> Thank you for pointing out this interesting direction. The exponential averaging version of your suggestion has been tried in De et al. 2022. Since our paper aims at gradient denoising, LP filter on the model for inference is out of the scope of the current paper, and we would like to leave it as a possible future direction.
>
> $\quad$
>
> Thank you for reading our rebuttal. We hope the above responses addresses your concerns, and if there are any remaining questions, we would be more than happy to continue discussing with you.

---

> > ### Author Response · Authors · 2024-08-07
> > **Additional comment to Reviewer vvxz**
> >
> > > Detailed comparison to correlated noise approaches for DP optimization
> >
> > We would like to provide a detailed discussion between our method and the correlated noise method  (KMST+, DMRST, and CDPG+) in this comment. As discussed in the above response, the update of correlated noise method can be written as $B(CG_{0:t}+W_{0:t}) = AG_{0:t}+BW_{0:t},$ where $B$ is a high-pass filter and our method is $AM(G_{0:t}+W_{0:t})$, where $M$ is a low-pass filter.
> >
> > **Connection:**
> >
> > The correlated noise methods and our proposed method can all be viewed as processing the signal in the frequency domain to "separate" the noise and the gradient.
> >
> > **Differences:**
> >
> > The existing correlated noise methods 1) pre-process the gradient/noise to separate the gradient and the DP noise in the frequency domain, and therefore require careful design of the matrices $B, C$ for each problem and optimizer, 2) require extra memory ($O(d\log(t))$ to $O(dt)$), which is unrealistic for large scale training, and 3) only work for SGD update, since Adam cannot be written as such a prefix sum of privatized gradient.
> >
> > In contrast, our method 1) post-processes the noisy signal to extract the gradient from the noise from the frequency domain, 2) only requires $O(d)$ extra memory, which is *independent* of $t$, and 3) is compatible with any first-order optimizer since it just post-processes the gradient.

---

> > > ### Comment · Reviewer_vvxz · 2024-08-09
> > > **Thanks for the response**
> > >
> > > Thank you for the detailed response! I have updated my score.
> > >
> > >
> > > I would still like to see empirical evidence for the autocorrelation assumption, potentially in toy problems.
> > > Right now, it looks too much like a trick for the math to go through without enough practical justification of the details.
> > >
> > >
> > > Some investigation on observed auto-correlation vs. choice of filter coefficients for real data would also be valuable.

---

> ### Author Response · Authors · 2024-08-09
> **Further response to Reviewer vvxz**
>
> Thank you for your timely response and for increasing your score! We sincerely appreciate the time you spent reading our response and providing additional feedback.
>
> We apologize for not elaborating further on our original response. We mistakenly assumed that you are asking about the power spectral density (PSD) plots, which are related to the auto-correlation coefficients and show the low-frequency property of the stochastic gradients in the frequency domain (as shown in Fig. 2 in the original manuscript and Fig. 4 in the PDF response). This was the reason we added Fig.4 in our rebuttal. The PSD plots are obtained by applying FFT to the auto-correlation coefficients. So, they are directly related to the auto-correlation plots. Having said that, it seems, unfortunately, we are unable to include more figures in the response, but by applying inverse FFT (iFFT) to the PSD, we can easily obtain the auto-correlation coefficients, and $c_\tau, c_{-\tau}.$ We will follow your suggestion and put the original auto-correlation coefficients directly showing $c_\tau$ and $c_{-\tau}$ in our revised paper.
>
> Regarding your comment on the investigation of the auto-correlation vs filter coefficients, a frequency-domain illustration is given in Fig. 2(b) in the original paper, which plots the auto-correlation of the stochastic gradients after applying the filter. By an inverse FFT on it, we can observe the relation between the coefficients of the low-pass filter and the resulting auto-correlation coefficients.
>
> To further address your question, we copy the first 10 coefficients of the auto-correlation of the stochastic gradients, and $\kappa_\tau$ of the filter:
>
> | Auto-correlation | 0.0436| 0.0273| 0.0194| 0.0151| 0.0131| 0.0058| 0.0026| 0.0021| 0.0005| 0.0004|
> | - | - | - | - | - | - | - | - | - | - | - |
> | Filter coefficients $\kappa_\tau$ | 0.0909| 0.1652| 0.1352| 0.1106| 0.0905| 0.0740| 0.0606| 0.0495| 0.0405| 0.0331|
>
> It can be observed that the auto-correlation and the filter coefficients are all gradually decreasing as $\tau$ increases. We would include these discussions in our revision. Please let us know if our response answers your question.
>
> Thank you again for your invaluable feedback.

---

> > ### Comment · Reviewer_vvxz · 2024-08-14
> > **Response**
> >
> > Thank you for the response and the additional details. I'm in favor of acceptance: I will maintain my score as I feel that it is a fair assessment of the paper.
> >
> > Some additional suggestions: Further justification of the autocorrelation assumption would be nice. Why is it always possible to choose $c_{-\tau}$ to be positive? I would have expected some sort of symmetry in $c_{\tau}$ and $c_{-\tau}$. Further details about this, and exploration for toy examples (e.g. simple quadratic functions or logistic regression for a slightly harder problem) would be quite helpful for a reader.

---

### Official Review · Reviewer_ahhw · 2024-07-07

**Soundness:** 2
**Presentation:** 4
**Contribution:** 3
**Rating:** 5
**Confidence:** 4

**Summary:**

This paper suggests the effects of a low-pass filter for private training with DP-SGD. After investigating the noise and true gradients during training, the authors propose using previous gradients and momentum to distinguish effective gradients from random noise. They empirically prove their idea across various settings, including different optimizers and datasets.

**Strengths:**

•	The paper presents a very simple and effective theory based on the optimization of DP-SGD and the similarities between gradients $\nabla F(x_t)$.

•	The authors investigate the optimization dynamics in terms of the frequency domain rather than the time domain.

•	The paper conducts experiments on various datasets and existing well-known optimizers, demonstrating the effectiveness of the frequency-aware optimization.

•	The authors establish an interesting relationship between SNR and frequency domain approaches.

**Weaknesses:**

Please refer to the Questions section.

**Questions:**

I will happily increase the score if the authors can address the following questions:

•	The authors use assumptions of bounded variance and gradient norm. However, these assumptions might not be true in real DP-SGD situations. The strong underlying assumption of the authors is that the whole-batch gradient directions are similar between timestamps $\nabla F(x_t)$ and $\nabla F(x_{t-\tau})$. This is widely known in GD settings, but it may not be obvious in SGD (and DP-SGD) settings. Can the authors provide evidence with mini-batch gradients $\nabla f(x_{t})$? [1,2,3]

•	Using a low-frequency signal typically requires the use of FFT and iFFT to distinguish patterns in the frequency domain. While I understand that the authors try to investigate the auto-correlation and PSD of gradients (as shown in Figure 1), I cannot agree that this approach is orthogonal to conventional approaches depending on timestamps. As the authors mentioned momentum, this approach seems somewhat similar to the learning dynamics of DP-SGD, rather than the frequency domain. The authors should clarify this issue.

•	There are some existing papers that investigated the use of previous gradients in DP-SGD that are missing from the discussion. [1] investigated the use of previous gradients and their momentum approaches for sharpness-aware training. They explored the correlation between previous private gradients and the current gradient during optimization.

•	For the momentum approaches, why don’t you use $\alpha$ and $1-\alpha$ for the popular setup in EMA? In the appendix, I saw that tuning both $\alpha$ and $\beta$ requires quite a large search space. I wonder if the effectiveness of your methods comes from the enlarged search space. Could you provide an ablation study for this search space?

•	The authors try to make a theoretical analysis (line 233), however, it may not be true in the real application. The learning rate of private training is much larger than standard training, where it cannot be lower than $O(\sqrt{1/\tau})$ as far as I know.

[1] Explicit loss asymptotics in the gradient descent training of neural networks, NeurIPS 21

[2] Measurements of Three-Level Hierarchical Structure in the Outliers in the Spectrum of Deepnet Hessians, ICML 18

[3] Implicit Jacobian regularization weighted with impurity of probability output, ICML 23

[4] Differentially Private Sharpness-Aware Training, ICML 23

**Limitations:**

The authors clarify the limitations.

---

> ### Author Rebuttal · Authors · 2024-08-07
>
> Thank you for recognizing the contribution of our paper and providing detailed feedback to improve the paper. Our responses to your specific comments are listed below.
>
> > The authors use assumptions of bounded variance and gradient norm. However, these assumptions might not be true in real DP-SGD situations. The strong underlying assumption of the authors is that the whole-batch gradient directions are similar between timestamps ...
>
> Let us review our main assumptions:
> Assumptions A.2 and A.3 require bounded gradient and bounded variance. These assumptions are standard for non-convex optimization and for DPSGD analysis, as discussed at the end of Sec. 2.1 of the manuscript. Specifically, the bounded variance assumption is standard for non-convex optimization. Notice that as long as the input is finite (e.g. pixels with finite RGB values), the variance is guaranteed to be bounded. The bounded gradient assumption is widely used in the analysis of DPSGD, and when the model parameters have a finite value, the gradient is always bounded.
> As you pointed out, A.4 is another major assumption that requires certain properties on the auto-correlation of the gradient. Notice that this assumption is on the gradient itself (and not noisy/stochastic versions). We provided a theoretical justification in the  *worst-case*, in Sec. 3 and lines 232-235, that when the learning rate is sufficiently small, A.4 always holds. Moreover, we have provided the empirical verification in Fig. 2, where the mini-batch gradients are correlated. We provided more figures illustrating the auto-correlation of the mini-batch gradient in the attached PDF. Please let us know if this addresses your concern.
>
> > Using a low-frequency signal ... I cannot agree that this approach is orthogonal to conventional approaches depending on timestamps...
>
> We believe there is some misunderstanding about our claim of *orthogonal to noise reduction method in the time domain*. First, we agree with the reviewer that our approach is *not* orthogonal to the momentum method, and other possible methods that depend on timestamps, as we discussed its connection with the momentum method in Sec. 4, lines 218-221, Sec. 5.3, lines 265-268.
> Instead, our approach **is orthogonal** to the approaches that do not rely on timestamps, e.g., modifying clipping operation, changing model structures, and using noise scheduler. These approaches aims at reducing the impact of the DP noise for each step independent of other steps and DOPPLER can be combine with these approaches. We will further clarify this point in our revision.
>
> > There are some existing papers that investigated the use of previous gradients in DP-SGD that are missing from the discussion...
>
> Thank you for bringing up this relevant literature. We will cite these papers and discuss them in our revised version. Notice that the philosophy and motivation behind DP-SAT and DOPPLER are different: DP-SAT aims at finding a "flat" minimizer. Moreover, their approach is different. In particular, the momentum used in DP-SAT is only for an estimation of the perturbation direction for SAM, which is not used for DP noise reduction. In contrast, DOPPLER aims to denoise the gradient using the past gradient with a low-pass filter.
>
>
> > For the momentum approaches, why don’t you use $\alpha$ and $1-\alpha$ for the popular setup in EMA?   In the appendix, I saw that tuning both $\alpha$ and  $\beta$ requires quite a large search space. I wonder if the effectiveness of your methods comes from the enlarged search space. Could you provide an ablation study for this search space?
>
> Thank you for this nice and critical comment. We would like to clarify how the filters are designed:
>  1. As discussed in Sec. 4 and 5.3, Momentum-SGD (choose $\alpha, 1-\alpha$) is a special case of the low-pass filter. However, such a choice might not achieve the best performance. Therefore, filters with more coefficients are needed, which admits a larger search space and possibly better performance.
>  2. The effectiveness of the method indeed comes from the enlarged search space and the frequency-domain viewpoint to efficiently design the filter coefficients. As discussed in the general response, there exists a series of mature methods to choose the filter coefficient.
>  3. In Fig. 6 in the original paper, we provide an ablation study to the choice of filter coefficients, and more choice of filter coefficients are provided in the PDF response.
>
> > The authors try to make a theoretical analysis (line 233), however, it may not be true in the real application. The learning rate of private training is much larger than standard training, where it cannot be lower than $O(\sqrt{1/\tau})$ as far as I know.
>
> Thank you for your detailed feedback on our paper. In our paper, the requirement of the learning rate is $\eta = O(\sqrt{1/\tau})$, which ensures the auto-correlation to be positive in the **worst case**. This requirement aligns with what the reviewer has suggested. Moreover, in real-world applications, the gradient can still be positively correlated for larger values of the learning rate. For example, for quadratic problem $1/2\|Ax+b\|^2$, by choosing $\eta \leq 1/\|A^\top A\|,$ it is guaranteed that the grdients are positively correlated. This does not invalidate our result as we only provide a bound.
> Moreover, as illustrated in Fig. 2 (blue line), the PSD of the stochastic gradient is low-frequency, indicating the positive correlation of the stochastic gradients in real-world applications. Similar observations were also made in other papers, e.g., [DPDR] Liu et al. 2024, Fig. 1.
>
> $\quad$
>
> Finally, we would like to thank you for reading our rebuttal and providing detailed feedback to us. We did our best to respond to all your comments. Please let us know if there are still any remaining questions and we would be more than happy to continue the discussion.

---

> > ### Comment · Reviewer_ahhw · 2024-08-11
> > **Thanks for the response**
> >
> > I carefully read the rebuttal of the authors, including the general response and answers to my questions and those of the other reviewers. Although I thought some of the mathematical support was still limited, I understand the authors' novelty points and the experimental results of the proposed method. Personally, the authors effectively addressed the questions, including mine and those of the other reviewers.
> >
> >  Thus, I have raised the score.

---

> > > ### Author Response · Authors · 2024-08-12
> > >
> > > Thank you for your kind response and for increasing your score! We sincerely appreciate the time you spent reading our response. Please let us know if you have any further comments and suggestions, and we would be more than happy to include them in our revised manuscript.

---

### Official Review · Reviewer_kR6J · 2024-07-13

**Soundness:** 3
**Presentation:** 2
**Contribution:** 2
**Rating:** 5
**Confidence:** 3

**Summary:**

This paper proposes DOPPLER mechanism to post-process DP gradients and reduce noise in them, before updating the model. The work makes an observation that, in frequency domain, there is a clear distinction between distribution of SGD gradients and DP noise; the earlier lies in a small window of lower frequencies while the latter is distributed uniformly over all frequencies. Based on this observation, the DOPPLER mechanism uses a low-pass frequency filters that filters all high frequency signals, and hence, most noise is filtered while only some of the useful gradient is filtered thereby increasing overall SNR. Paper presents theoretical connection between SGD gradients and DP noise in the time and frequency domains, privacy analysis and empirical evaluation on standard benchmarks to showcase efficacy of DOPPLER.

**Strengths:**

- Interesting signal processing perspective for DP optimization problem and observation about gradients and noise in frequency domain
- DOPPLER is a post-processing method, hence can be used with SOTA DP methods
- Experimental results show that in training-from-scratch settings DOPPLER is useful

**Weaknesses:**

- Connection between time/frequency domains is difficult to understand; adding some background might be useful
- Motivation of the paper is to enable DP training for large, foundation models, but experiments are performed on relatively small models
- DOPPLER can be combined with any SOTA DP training, but results with such SOTA methods are missing

**Questions:**

Section 3:
- Can you provide some intuition about what does it mean to convert a series of gradients from time to frequency domain? It might be useful to have this somewhere (even if it’s in appendix) to help readers understand the approach better.
- Line 167-168: In Figure 1a, auto-correlation coefficient first increases and then decreases with time, but description seems to state something else. Can you clarify?
- Line 168-169: How do you compute PSD? How do you go from the first equation in Section 3 to drawing Figure 1b, 1c?
- Line 187-188: Why can a linear low pass filter be written as in the first equation of section 3.1?

Section 6:
- What are the sizes of the models used?
- Why do you not compare with SOTA methods, e.g., De et al. 2022 or Shejwalkar et al. 2022?

Shejwalkar et al., Recycling Scraps: Improving Private Learning by Leveraging Intermediate Checkpoints, arXiv 2022

**Limitations:**

- Proposed method, although motivated from enabling DP for large models, cannot be used for large models due to computation inefficiency.
- Current set of results lacks depth and should be improved.
- Paper writing/clarity can be improved (see questions/weaknesses).
  - Equations are not properly numbered.

---

> ### Author Rebuttal · Authors · 2024-08-07
>
> Thank you for recognizing the contribution of our signal processing perspective, and the precious advice for improving the presentation. We would like to address the reviewer's concerns as follows.
> $\quad$
> > Can you provide some intuition about what does it mean to convert a series of gradients from time to frequency domain? It might be useful to have this somewhere (even if it's in appendix) to help readers understand the approach better.
>
> Great suggestion! We agree that the paper can benefit from further discussion of the background. Please take a look at our proposed new appendix section in our general response to all reviewers and let us know if you would like to see further discussions on any specific background. The intuition of converting the gradient sequence in the time domain to the frequency domain is that the frequency domain representation helps us to capture and utilize certain properties of the gradient that are hard to observe in the time domain, e.g., the separability of the noise and gradient as discussed in Sec. 3.
>
> > Line 167-168: In Figure 1a, auto-correlation coefficient first increases and then decreases with time, but description seems to state something else. Can you clarify?
>
> Let us clarify that in Fig. 1(a) (also in 1b) and 1c)), we shift the x-axis so that $\tau = 0$ ($\nu = 0$) is in the middle of the plots, because the auto-correlation coefficients are symmetric, i.e., $\mathbb{E}[\nabla F(x_t)^\top\nabla F(x_{t+\tau})] = \mathbb{E}[\nabla F(x_t)^\top\nabla F(x_{t-\tau})].$ Therefore, the auto-correlation coefficients decreases as the time-lag $|\tau|$ increases, instead of first increase then decrease.
>
> > Line 168-169: How do you compute PSD? How do you go from the first equation in Section 3 to drawing Figure 1b, 1c?
>
> As explained in lines 162-163 in the original manuscript, the PSD of the gradient is computed by applying a  Fourier transform to the auto-correlation coefficients in figure 1(a), i.e., $P(\nu) = \mathcal{F}\{\phi(\tau)\}.$ The explicit transform can be found in the general response.
>
> > Line 187-188: Why can a linear low pass filter be written as in the first equation of section 3.1?
>
> A linear filter is a filter whose output is a **linear combination** of the past input signals, which can be written in the general form of $$m_t = -\sum_{\tau=1}^{n_a}a_\tau m_{t-\tau} + \sum_{\tau=0}^{n_b}b_\tau g_{t-\tau}.$$ The choice of the filter coefficients $\{a_\tau\}, \{b_\tau\}$ determine whether the filter is a high-pass, low-pass, band-pass or band-stop filter. This should be clear after we add the suggested new appendix on the background.
>
> > What are the sizes of the models used?
>
> In the experiments, we report the results on a 5-layer CNN (223K), ViT-small (30.1M), EfficientNet (5.33M), and ResNet-50 (25.6M) for the CV experiments (in Appendix B.4 in the original manuscript), and a RoBERTa-base model with 125M parameters. Note that for DP pre-training, these models are considered as **large models** compared with the ones used in the existing literature.
>
> > Why do you not compare with SOTA methods, e.g., De et al. 2022 or Shejwalkar et al. 2022?
>
> We would like to clarify that the SOTA methods, De et al. 2022 and Shejwalkar et al. 2022, implement a series of engineering tricks to improve the performance of DP training, as discussed in Sec 2.3 in the paper. We implemented several methods proposed by these papers in our experiments, including model design, group normalization, bounded activation, and large batch size.
> However, the rest of the engineering tricks (in JAX) are not memory/time efficient and are incompatible with our implementation in PyTorch. Therefore, although we did not directly compare these SOTA methods with their DOPPLER version, our numerical experiments *implicitly* adopt part of these SOTA methods in our comparison.
>
> > Proposed method, although motivated from enabling DP for large models, cannot be used for large models due to computation inefficiency.
>
> We believe that the reviewer's comment on computation inefficiency is not entirely accurate. We agree with the reviewer that the proposed method requires more memory than the standard DP optimizer.
> However, the computational cost of applying the Low-pass filter is *at the same level* as momentum-SGD, with an overhead of combining the past gradients. This cost is negligible compared with the backward and clipping steps. For example, when using DPSGD to train a ResNet with Cifar-10, the backward and clipping takes 16240ms (406ms/50 samples) for one minibatch (2000 samples), while the SGD update takes 52ms, and for LP-DPSGD, the Low-pass filter only takes an extra 74ms, i.e., $\sim0.5\\%$ overhead in computation.
>
> > Current set of results lacks depth and should be improved.
>
> We believe our current results covers a wide range of algorithms, models, and datasets. We also provide more numerical results in the PDF response for in-depth investigation on the filter design. However, we understand that this is a bit subjective matter and readers may expect different/additional experiments. We would gladly add more if you could please tell us what is missing in the experiments, and how we can address it.
>
> > Paper writing/clarity can be improved (see questions/weaknesses). Equations are not properly numbered.
>
> We agree with the reviewer and appreciate for the valuable feedback, and we will an additional section to the appendix covering the background. Also, we will revise the main manuscript as the reviewer suggested to clarify the points. We will re-label the equations. However, we believe that in the current manuscript, all referred equations have been labeled.
>
> $\quad$
>
> Thank you for reading our rebuttal. We hope the above responses have addressed your concerns.If there are still any questions, we would be more than happy to continue discussing with you.

---

> > ### Author Response · Authors · 2024-08-14
> >
> > Dear reviewer,
> >
> > We would like to thank you again and kindly remind you that the discussion period will end soon. We hope our response has addressed your concerns. Thank you very much for the time you spent reviewing our work and providing constructive feedback to us.

---

### Author Rebuttal · Authors · 2024-08-07

# Response to all reviewers
We would like to thank the reviewers for their constructive and detailed feedback. We are glad that the reviewers found our approach novel and  effective despite its simplicity. Before responding to the individual reviewers' questions, we would like to thank the reviewers for their suggestion on adding a discussion on the signal processing background. We completely agree that such a discussion would improve the paper. Therefore, in our revised version, we plan to include an additional appendix discussing the background. Below is our suggested appendix on the background:

## Background in signal processing
### Frequency domain analysis
* **What is frequency domain analysis.** In signal processing, frequency domain analysis is used to analyze the periodical or long-term behavior of a  (time series) signal/data. In the frequency domain analysis, we use the frequency $\nu$ as the indices of the signal, e.g., $\{X(\nu)\}, X(\nu)\in \mathbb{C}$, where each term $X(\nu)$ records the amplitude and phase of the sine wave of frequency $\nu$ that composes the signal; in contrast, in time domain, we use time $t$ as the indices of a signal, e.g., $\{x_t\}$, where each term $x_t$ records the value of the signal at a given time $t$.
In the paper, we treat each coordinate $i \in [1, \dots, d]$ of the privatized gradient over the iterates as an individual signal, i.e. $\{g_1[i], g_2[i], \dots, g_T[i]\}$. Thus the gradient over iterates gives us $d$ one-dimensional signals and we can look at their frequency domain representation of each signal.
* **Why converting a signal to the frequency domain is beneficial.**
    1) Certain properties of a signal can be hard to observe/characterize in the time domain. For example, a long-time correlation or a cyclic behavior of the signal is not easy to directly observe in the time domain. By converting the signal to the frequency domain, such properties can easily be captured and analyzed. For example, the signal $x_t = \sin( t)$ has nonzero entries in almost all times. However,  the frequency domain representation of this signal has only one entry that is non-zero, i.e., $X(1) = 1$ and all other entries are zero, i.e., $X(\nu) = 0, \forall \nu \neq 1.$ This means  $x_t$ has only one periodic signal in it.
    2) Certain mathematical analysis can be significantly simplified in the frequency domain. For examples, linear differential equations in the time domain are converted to algebratic equations in the frequency domain; filters as convolutions in the time domain are converted to point-wise multiplication in the frequency domain. These properties greatly simplifies the analysis of the signals and filters' dynamic. See Sec. 3.7, 10.5 in Oppenheim et al., 1996 for detailed discussion.
* **How to obtain a frequency domain representation of a signal.** To obtain a frequency domain representation of a discrete signal, one can apply Discrete Fourier transform (DFT) ($\mathcal{F}\{x_t\}: X(\nu) = \sum^{T-1}_{t=0} x(t)e^{\frac{-2\pi i t}{T}\nu}$)) to the signal. By directly applying DFT to a signal and obtaining $\{X(\nu)\}$, one can identify how the signal is composed of sin waves of different frequencies $\nu$ with their amplitudes and phases. In the paper, we apply DFT to the auto-correlation of a signal and obtain its power spectrum density (PSD). The PSD of a signal shows the distribution of the power of a signal on different frequencies. For example, the PSD of $x(t) = \sin(t)$ is $P(\nu) = 1/2$ for $\nu = \pm\frac{1}{2\pi}$ and 0 elsewhere.

### Low-pass filter
* **Frequency filter.**  A  frequency filter is a transformation of a signal that only allows certain frequencies to pass and blocks/attenuates the remaining frequencies. For example, for a signal $x(t) = \sin(t) + \sin(10t),$ we can apply an (ideal)  low-pass filter $F(\nu) = 1$ when $|\nu| \leq \frac{1}{2\pi}$ and $0$ otherwise. Then after applying the filter, $F*x(t) = \sin(t)$, the output signal only keeps the low-frequency signal.
* In this work, we use (time-invariant) linear filters for DP noise reduction. A linear filter attenuates certain frequencies by using a linear combinations of the input signal. Considering $g_t$ as the time signal, the general form of a linear filter on $g_t$ is $$m_t = \sum_{\tau=0}^t \kappa_\tau g_{t-\tau} = -\sum_{\tau=1}^{n_a}a_\tau m_{t-\tau} + \sum_{\tau=0}^{n_b}b_\tau g_{t-\tau},$$ where $\kappa_\tau$ are the filter coefficients. The second formula is a recursive way of writing the filter.

* **Filter design.** The property of the filter depends on the choice of the filter coefficients. Designing a filter consists of the following steps:
    1. Decide filter order/tab $n_a, n_b$. Larger $n_a, n_b$ give the filter more flexibility and better possible performance, at a cost of more memory consumption. In our experiment, we tested on 0th-3rd order filters, i.e., $\max\{n_a, n_b\} \leq 3$.
    2. Decide filter coefficients $\{a_\tau\}, \{b_\tau\}$.  Filter design can in general be a complex procedure and it involves deciding on trade-offs among different properties of the filter. Two standard constraints on the filter coefficients are: a) $-\sum a_\tau + \sum b_\tau = 1$, to ensure the filter has unit gain, i.e., the mean of the signal remains unchanged; and b) the solutions $x$ to $1 + \sum a_\tau x^\tau = 0$ satisfies $|x|<1,$ to ensure the filter is stable, i.e., $\sum|\kappa_t| < \infty.$ In the paper, we directly follow the design of Chebyshev filter and Butterworth filter, and tuning their cut-off frequency (and ripple) to achieve the best performance and maintaining these properties. See Winder, 2002 for detailed discussion.
* **Plot of the frequency response of the filters.** In the PDF response, we provide the time response ($\kappa_\tau$) and frequency response of the filters used in Tab. 2 in the original paper and additional experiments.

---

### Decision · Program_Chairs · 2024-09-25

**Decision:**

Accept (poster)

**Comment:**

The reviewers were overall positive about the paper. There were some valuable concerns raised during the rebuttal period, especially by reviewer vvxz, in regards to the autocorrelation assumption. The authors diligently addressed most of the concerns raised. We would recommend the authors to include most or all of the discussion from the rebuttal in the next iteration of the paper.